# PERMANENT AND TRANSIENT REPRESENTATIONS FOR CONTINUAL REINFORCEMENT LEARNING

## ABSTRACT

Continual Reinforcement Learning agents struggle to adapt to new situations while retaining past knowledge, resulting in a stability–plasticity trade-off. An appealing solution is to decompose the agent's predictions into permanent and transient components—one for long-term retention and the other for rapid adaptation—thereby achieving a better balance (Anand & Precup, 2023). Building on this idea, we propose using different sets of feature representations to estimate permanent and transient value functions, enabling even faster adaptation. We demonstrate the effectiveness of our approach on small-scale examples for both prediction and control tasks, analyze its theoretical properties, and show its benefits on the Craftax-Classic benchmark using a novel non-parametric approximator for transient value function estimation. Our method facilitates online learning and outperforms the PQN baseline.

## 1 INTRODUCTION

Continual reinforcement learning (CRL) is a key ingredient for understanding intelligence and building agents that autonomously adapt to changes in their environment (Sutton et al., 2022; Silver & Sutton, 2025). An important challenge for artificial CRL agents is the tension between retaining knowledge already acquired while adapting to new information—the *stability–plasticity* trade-off (Carpenter & Grossberg, 1987). In contrast, humans and other natural intelligences adapt to changes in their environment throughout their lifetime. Kumaran et al. (2016) posited that this natural ability is due to the existence of two complementary learning systems (CLS): one that adapts rapidly and another that slowly consolidates information across experiences. Inspired by CLS theory, Anand & Precup (2023) introduced a decomposition of the value function into a permanent component, which provides a stable baseline estimate for any situation the agent may face, and a transient component, which adapts these estimates to the present context by applying temporal-difference corrections. This approach led to improved performance in both prediction and control problems with various forms of value function approximation. However, they use the same features for both value functions. Intuitively, using separate feature spaces for these approximators would be more aligned with the idea of keeping them complementary: one system should compensate for the weaknesses of the other, making the framework both more effective and more biologically plausible. For example, permanent features should intuitively encode *stationary or slowly changing* components of the environment, such as the map of a city, while transient features can capture *situation-specific* aspects, such as a road being blocked on a given day.

In this paper, we develop this idea, establish some theoretical results and provide empirical evidence that it scales to large environments. For this purpose, we develop a novel non-parametric approximator that operates directly on raw observations and can be used to learn efficiently a transient representation and value function. Its design combines the strengths of tabular learning, CMACs (Miller et al., 1990), and tile coding (Sutton, 1995), enabling precise corrections at a rapid pace, facilitating online learning, and providing controlled generalization—ultimately leading to faster CRL.

Our main contributions are as follows:

- We build on the permanent–transient value function framework to incorporating separate feature representations;

- We establish convergence guarantees under linear function approximation for both permanent and transient value functions using a two-timescale convergence technique;
- We demonstrate the effectiveness of the approach in both prediction and control through small-scale experiments;
- We introduce a novel non-parametric method for representing transient features, which can be used to complement neural network-based permanent features;
- We evaluate our design in online CRL on the 18M and the 250M Craftax benchmark, showing favourable comparisons with a strong PQN baseline.

## 2 BACKGROUND

CRL agents exhibit *endless adaptation*, as opposed to converging to a fixed solution (Abel et al., 2023; Sutton et al., 2007). The need for CRL arises when aspects of the environment—such as rewards or transition dynamics—change over time (Khetarpal et al., 2022; Pan et al., 2025), or when the agent's resources are limited relative to the complexity of its environment, thereby creating the need for the agent to keep updating its limited knowledge (Kumar et al., 2025; Javed & Sutton, 2024; Lewandowski et al., 2025). In such scenarios, the agent must balance retaining useful information from the past to adapt more quickly when similar situations reappear in the future—*stability*—with allocating resources to learn from new experiences—*plasticity*.

Because neural networks have become the main approach to approximating value functions and policies (Mnih et al., 2013), most CRL research has focused on understanding the stability–plasticity trade-off in neural networks (Lyle et al., 2022; Nikishin et al., 2022; Abbas et al., 2023; Lyle et al., 2023; 2024) and on developing new regularization techniques (Lewandowski et al., 2024; Chung et al., 2024) and optimizers (Kirkpatrick et al., 2017; Jones et al., 2022; Dohare et al., 2024) to improve it.

Building on earlier ideas of decomposing value functions in model-based RL (Silver et al., 2008), Anand & Precup (2023) proposed splitting both the value function and the action-value function into two components to trade-off stability and plasticity: permanent components, $V^{(P)}$ and $Q^{(P)}$, which learn general estimates from the entire agent experience (similar to Abel et al. (2018) in transfer learning), and transient components, $V^{(T)}$ and $Q^{(T)}$, which adapt these estimates to the current situation. The overall value functions, $V^{(PT)}$ and $Q^{(PT)}$, are then expressed as sums of these two components:

$$V^{(PT)}(s) = V_{\mathbf{w}}^{(T)}(s) + V_{\theta}^{(P)}(s), \tag{1}$$

$$Q^{(PT)}(s,a) = Q_{\mathbf{w}}^{(T)}(s,a) + Q_{\theta}^{(P)}(s,a), \tag{2}$$

where $\theta$ and $\mathbf{w}$ are the parameters of the permanent and transient function approximators, respectively.

The permanent value function is updated more slowly, in phases –either every $k$ timesteps or at task boundaries (when available)– by using experience from that phase:

$$\theta_{k+1} \leftarrow \theta_k + \overline{\alpha}_k \left( V^{(PT)}(S_k) - V_{\theta}^{(P)}(S_k) \right) \nabla_{\theta} V_{\theta}^{(P)}(S_k), \tag{3}$$

$$\theta_{k+1} \leftarrow \theta_k + \overline{\alpha}_k \left( Q^{(PT)}(S_k, A_k) - Q_{\theta}^{(P)}(S_k, A_k) \right) \nabla_{\theta} Q_{\theta}^{(P)}(S_k, A_k), \tag{4}$$

where $\overline{\alpha}$ is the learning rate for permanent updates.

In contrast, the transient value function updates rapidly to capture aspects of the value that are not yet reflected in the permanent estimates:

$$\mathbf{w}_{t+1} \leftarrow \mathbf{w}_t + \alpha_t \left( R_{t+1} + \gamma V^{(PT)}(S_{t+1}) - V^{(PT)}(S_t) \right) \nabla_{\mathbf{w}} V_{\mathbf{w}}^{(T)}(S_t), \tag{5}$$

$$\mathbf{w}_{t+1} \leftarrow \mathbf{w}_t + \alpha_t \left( R_{t+1} + \gamma \max_{a'} Q^{(PT)}(S_{t+1}, a') - Q^{(PT)}(S_t, A_t) \right) \nabla_{\mathbf{w}} Q_{\mathbf{w}}^{(T)}(S_t, A_t), \tag{6}$$

where $\alpha$ is the learning rate for transient updates. To maintain plasticity, transient parameters are decayed or reset after each permanent update.

## 3    PERMANENT AND TRANSIENT REPRESENTATIONS

Since the permanent and transient and value functions should intuitively complement each other, it is reasonable to imagine them using distinct representations, with different requirements. In CLS (Kumaran et al., 2016), the equivalent of the transient representation is considered to be the hippocampus, which stores latent embeddings of recent trajectories. These are later replayed during sleep, resulting in consolidation and slow learning of a permanent representation, thought to be located mainly in the prefrontal cortex. This representation provides good generalization to new situations and supports long-term planning.

In the rest of the paper, we consider architectural choices which implement this intuition for CRL agents. Specifically, the permanent representation should be expressive enough to learn baseline predictions for any situation the agent might encounter. Learning can be slow, but the information acquired should persist and be useful for a long period of time (ie. high stability). Therefore, the permanent representation should support broad generalization of predictions between similar situations. Many *neural network architectures*, such as feedforward, convolutional, or recurrent, meet these desiderata, making them good candidates for the permanent value function.

The transient representation should support online learning at a rapid pace (ie. high plasticity), in order to adapt quickly to new situations. Moreover, if we assume that the agent's circumstances can change rapidly, the transient representation should facilitate learning precise estimates, with minimal or carefully controlled generalization around the current data. Additionally, the transient representations should allow fast and accurate information retrieval. As in CLS, knowledge stored in the transient representation should support long-term learning of the permanent representation.

In the following sections, we present both theoretical and empirical analysis of simpler architectures based on these intuitions. Then, in Sections 6 and 7, we develop and test a new approach to implementing the transient memory, which respects the goals above while providing better generalization than simple replay buffers.

## 4    THEORETICAL RESULTS

In this section, we study the convergence of permanent and transient value function updates when these are based on different feature spaces, by leveraging the two-timescale proof technique pioneered by Borkar (see Appendix 2) (Borkar, 1997). While his approach is general and broadly applicable, the conditions can be simplified in the context of RL, as shown by Bertsekas & Tsitsiklis (1996); Tsitsiklis & Van Roy (1996) for a single iteration. In particular, the Lipschitz assumption is satisfied by showing that the expected update in matrix form (ie. the key matrix) is well-defined and positive definite. His final assumption is satisfied by first showing that the noise terms form *martingale difference sequence* with zero mean and bounded variance. In our analysis, we consider updating permanent value function at each timestep.

We make the following assumptions[1]:

**Assumption 1.** The step-sizes $\overline{\alpha}$ and $\alpha$ satisfy $\sum_t \overline{\alpha}_t = \infty$, $\sum_t \overline{\alpha}_t^2 < \infty$, $\sum_t \alpha_t = \infty$, $\sum_t \alpha_t^2 < \infty$, $\lim_{t \to \infty} \frac{\overline{\alpha}_t}{\alpha_t} \to 0$.

**Assumption 2.** The permanent and transient feature matrices, $\Phi \in \mathbb{R}^{|\mathcal{S}| \times d}$ and $Z \in \mathbb{R}^{|\mathcal{S}| \times l}$, are full column rank, ie. the column vectors are linearly independent. Also, their norms are bounded, $\|\Phi\| \leq M_1$, $\|Z\| \leq M_2$ where $M_1$ and $M_2$ are constants.

**Assumption 3.** There are $N$ tasks and task $\tau$ is i.i.d. sampled according to $p_\tau$. Each task is an MDP, $\mathcal{M}_\tau = (\mathcal{S}, \mathcal{A}, \mathcal{R}_\tau, \mathcal{P}_\tau, \gamma)$, let $\mathbb{E}_\tau$ denote the expectation with respect to the task distribution. The rewards for each task $\mathcal{R}_\tau$ are bounded. The task boundaries are observable.

**Assumption 4.** Every task, $\tau$, induces irreducible, aperiodic Markov chain under the fixed evaluation policy $\pi$ and the chain is rapidly mixing:

$$|\mathcal{P}_{\pi,\tau}(S_t = s|S_0) - d_\pi(s)| \leq C\sigma^t, \ \forall S_0 \in \mathcal{S}, \sigma < 1,$$

where $C$ is a constant.

---

[1]We use the notations defined here in the proofs presented in Appendix.

**Theorem 1** (Main result). *Under Assumptions 1–4, the sequence of expected updates computed by permanent and transient updates converge to a unique fixed point:*

$$\theta^* = \mathbb{E}_\tau[\Phi^T D_\tau \Pi_{Z,\tau} \Phi]^{-1} \mathbb{E}_\tau[\Phi^T D_\tau Z \mathbf{w}_{Z,\tau}^{(TD)}],$$

$$\mathbf{w}_\tau^* = \mathbf{w}_{Z,\tau}^{(TD)} - \left(Z^T D_\tau (I - \gamma \mathcal{P}_{\pi,\tau}) Z\right)^{-1} Z^T D_\tau (I - \gamma \mathcal{P}_{\pi,\tau}) \Phi \theta^*,$$

*where*

$$\Pi_{Z,\tau} = Z \left(Z^T D_\tau (I - \gamma \mathcal{P}_{\pi,\tau}) Z\right)^{-1} Z^T D_\tau (I - \gamma \mathcal{P}_{\pi,\tau}),$$

$$\mathbf{w}_{Z,\tau}^{(TD)} = Z \left(Z^T D_\tau (I - \gamma \mathcal{P}_{\pi,\tau}) Z\right)^{-1} Z^T D_\tau \mathcal{R}_{\pi,\tau}.$$

*Proof.* We outline the proof here, with full details provided in Appendix.

Because of the disparity in learning rates between permanent and transient updates, the permanent estimates appear stationary while the transient values are being updated. And, the transient values appear converged when analyzing permanent updates.

We first establish the convergence of the transient updates, treating the permanent values as fixed, by verifying the conditions outlined in Theorem 3 (Tsitsiklis & Van Roy, 1996) (see Lemma 1 in Appendix). By substituting the fixed point of the transient parameters into the permanent updates, we then show that the required conditions are satisfied for them as well, and therefore convergence follows from Theorem 2 (Borkar, 1997) (see Lemma 2 in Appendix). □

**Corollary 1.** *If $Z = \Phi$, then*

$$\theta^* = \mathbb{E}_\tau\left[\Phi^T D_\tau \Phi\right]^{-1} \mathbb{E}_\tau\left[\Phi^T D_\tau \Phi \mathbf{w}_{\Phi,\tau}^{(TD)}\right],$$

$$\mathbf{w}_\tau^* = \mathbf{w}_{\Phi,\tau}^{(TD)} - \theta^*.$$

*Moreover, in the single-task setting, $\theta^* = \mathbf{w}_{\Phi,\tau}^{(TD)}$ and $\mathbf{w}_\tau^* = 0$.*

*Proof.* The proof is included in Appendix 2. □

The above corollary implies that in the single-task setting, if both the permanent and transient value functions are approximated using the same feature representation, then the permanent value function alone suffices to capture the predictions, while the transient component converges to zero.

## 5 SMALL-SCALE EXPERIMENTS

We conducted experiments on both prediction and control problems, where the value function and action-value function were estimated using a linear function approximator. In these experiments, we assume that the agent's experience can be divided into tasks, and that the task boundaries are known to the agent (semi-continual RL). The transition dynamics remain fixed across tasks, while the reward function changes. Through these experiments, we show that using separate features to approximate permanent and transient value functions results in faster adaptation in CRL. The pseudocode is provided in Appendix 1 and 2 and the details of hyperparameter sweeps in Appendix A.2.

### 5.1 PREDICTION

For the prediction problem, we use the $5 \times 5$ discrete gridworld environment shown in Fig. 1. The agent starts in the central state and can choose from four navigation actions, one for each cardinal direction. Each action typically moves the agent to the adjacent state, but the intended action is replaced by one of the two perpendicular actions with $10\%$ probability. The agent receives a reward when it transitions into a designated goal state, located in one of the corners highlighted in green; otherwise, no reward is given. Rewards are modified across tasks to introduce non-stationarity, as described in Table 1 in Appendix.

We use a Fourier basis (Konidaris et al., 2011) up to second order to approximate the value function. In our variant of PT-TD learning, second-order features are used to approximate the transient value function, while first-order features are used to approximate the permanent value function. The original PT-TD learning method (denoted as NeurIPS) uses all features for approximating both permanent and transient value functions. In both variants, the transient weights are reset to zero at the beginning of each task to induce plasticity, while previously learned values are retained through the permanent weights.

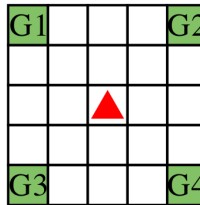

Figure 1: Grid task.

For comparison, we include two TD-learning baselines on which PT-TD learning is built. In the reset variant, all function-approximator weights are reset to zero at the start of each new task (full plasticity). In the continual variant, the agent continually updates its estimates on top of previously learned approximations. All algorithms are hyperparameter tuned thoroughly as outlined in Appendix A.2.

To evaluate performance, we use a uniformly random policy with a discount factor of 0.9. Each experiment runs for 1000 episodes, with the task changing every 100 episodes. The agent must therefore continually update its estimates to adapt to the current task. We use root mean squared value error (RMSVE) as a performance metric (the lower, the better). We report the mean and $90\%$ confidence intervals over 30 random seeds, computed with $z^* = 1.645$.

**Results:** The results are shown in Fig. 2a. In our variant, higher-order terms are used only for estimating the transient value function, enabling quick, low-variance, and precise adjustments to the permanent value function, resulting in the lowest overall RMSVE. First-order features provide sufficient expressivity while remaining low variance, making them well-suited for approximating the permanent value function (see Figure 9). The NeurIPS variant has slightly higher RMSVE due to the increased variance from using all features to approximate both value functions. Both PT-TD learning variants retain prior knowledge through the permanent value function, leading to lower error at the onset of tasks that reappear. In contrast, the TD-learning baselines perform poorly, as they fail to preserve previous predictions while simultaneously adapting to new tasks, corroborating prior findings (Anand & Precup, 2023).

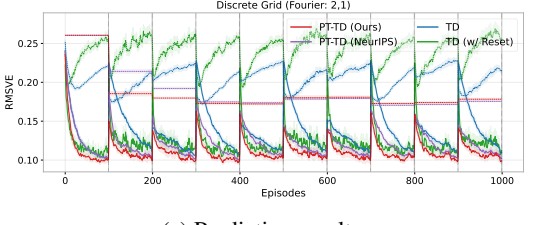
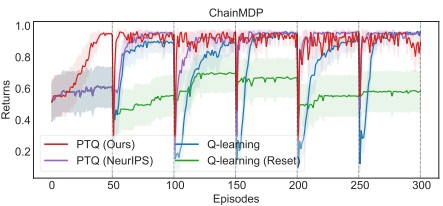

(a) Prediction results.

(b) Control results.

Figure 2: Prediction and control results in Semi-CRL. Our method (red line) achieves lower RMSVE in prediction and higher cumulative rewards in control compared to baseline algorithms.

## 5.2 CONTROL

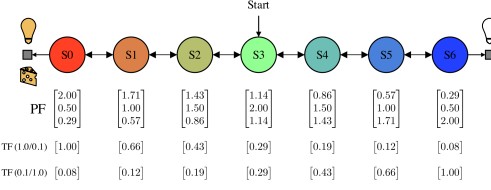

Figure 3: Chain task.

For the control problem, we use the simple chain environment with seven states shown in Fig. 3. The agent starts in the central state and can choose either the *left* or *right* action to move to the neighbouring state in the corresponding direction, with a $10\%$ probability of a reversed effect. Goal states with positive rewards are located at both ends of the chain. Non-stationarity is introduced by varying the reward magnitude across tasks, as described in Table 2 in Appendix.

Each state is represented using four features: the first three correspond to the red, green, and blue (RGB) components of the state, while the fourth encodes the light intensity, which depends on which end of the chain contains the reward. In our variant of PTQ-learning, the first three features

are used to approximate the permanent value function, and the last feature is used for the transient value function. In contrast, all other algorithms we compare against—including the NeurIPS variant of PTQ-learning and the reset and non-reset variants of Q-learning—use all features to approximate the value function. For the reset variant of Q-learning, the function-approximator weights are reinitialized at the beginning of each task.

We use discount factor of 0.99 and compute episodic returns to compare various algorithms. We run the experiment for 300 episodes, with the task changing every 50 episodes. We report the mean and 90% confidence intervals over 30 random seeds, computed with $z^* = 1.645$.

**Results:** The results are shown in Fig. 2b. Because our approach uses a single reward-correlated feature to adjust transient value estimates, adaptation is rapid. It is the only method that learns a meaningful behaviour within the first 50 episodes and re-adjusts it within five episodes whenever tasks change. PTQ-learning with all features fails to learn a meaningful policy during the initial 50 episodes and adapts marginally more slowly when tasks subsequently change. The reset variant of Q-learning performs the worst, as no prior knowledge is retained at the start of each task. The continual variant of Q-learning requires more time to overcome bias in its action-value estimates and is therefore slower to adapt.

These experiments demonstrate that using separate features for the permanent and transient value functions results in quicker adaptation.

## 6 NON-PARAMETRIC TRANSIENT MEMORY

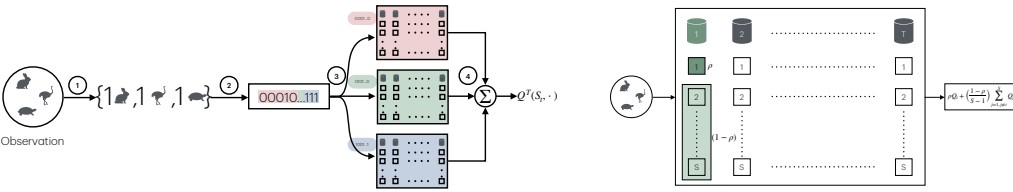

(a) An end-to-end overview of transient memory.      (b) Transient value table.

Figure 4: (a) Four-step process: (1) tokenization; (2) hashing; (3) binning; (4) value estimation. (b) Within each table, the value of the query observation is weighted by $\rho$, and the remaining mass, $(1 - \rho)$, is distributed among its neighbours to form the estimate.

In this section, we introduce a new non-parametric architecture that meets the desiderata of a transient memory (as described in Sec. 3), inspired by tile coding (Sutton, 1995) and CMACs (Miller et al., 1990).

We suppose that each observation consists of several channels, with each channel indicating the presence or absence of a specific item type in a particular location in the agent's view. Many commonly used RL environments, such as gym-Minigrid (Chevalier-Boisvert et al., 2023), Craftax (Matthews et al., 2024), and MinAtar (Young & Tian, 2019), are structured in this way. This assumption will allow us to produce hash signatures of observations using MinHash (Broder, 1997).

The overall process is illustrated in Fig. 4. It transforms an observation into a sparse, hashed representation that supports non-parametric value estimation with controlled generalization. The pipeline has four main stages:

**1. Tokenization:** Each observation is converted into a set of discrete tokens, where each token encodes the type of object present at a particular $(x, y)$ position in the agent's view. This converts the raw grid-like observation into a symbolic set representation.

**2. Hashing:** The tokenized set is mapped to a compact hash signature and an auxiliary tag using several independent MinHash functions. MinHash is used because it preserves the Jaccard similarity: the probability that two sets produce the same MinHash value is exactly their Jaccard score.[2]

---

[2]The Jaccard score between two sets $A$ and $B$ is, $J(A, B) = \frac{|A \cap B|}{|A \cup B|}$, and it is widely used in retrieval (Broder, 1997), bioinformatics (Ondov et al., 2016), and clustering (Tan et al., 2016).

As a result, observations that share many elements produce similar signatures (ie. overlap in many bits), which forms the basis for local generalization.

**3. Binning:** The signature is partitioned into multiple subsequences, each of which indexes into a bin in a transient table. Conceptually, each table corresponds to one *tiling* in tile coding (Sutton, 1995). Within a bin there are $S$ slots. When a new observation maps into a bin, it occupies an empty slot if available; otherwise, the least recently used (LRU) slot is evicted. The slot's initial value is set to the average of values already stored in that bin, so new entries inherit some local context. The auxiliary tag computed in the previous step is also stored in the slot, making it unique to that observation (see Alg. 3).

**4. Value estimation:** To induce generalization akin to CMACs (Miller et al., 1990), the value of a query observation is not taken from a single slot alone. Instead, its estimate is a convex combination of the focal slot and the other occupied slots in the *matched bin* (see Fig. 4b):

$$Q_{t+1}^{(T)}(x, a; k) = \rho Q_{t+1}^{(T)}(x, a; k) + \left(\frac{1-\rho}{S-1}\right) \sum_{j \neq i} Q_{t+1}^{(T)}(o_j, a; k), \tag{7}$$

where $\rho$ controls the relative weight of the focal slot versus its neighbours, $x$ is the query observation, $o_j$ is the observation stored in slot $j$ of the matched bin, $S$ is the total number of slots per bin, and $Q_{t+1}^{(T)}(x, a; k)$ is the transient value of the $(x, a)$ pair estimated by the $k$-th table. This encourages smooth interpolation between similar observations while still preserving distinct slot identities. The final transient value is obtained by summing the estimates from all transient tables. For more details, see 4 and 3 in Appendix.

Since several tables and their corresponding bins contribute to the overall estimate of the transient value function, the TD-error is distributed among them in proportion to their contributions (see 5 in Appendix). Specifically, the TD-error is first split evenly across all $T$ tables, and within each table it is further divided among the slots in the *matched bin*. The focal slot receives weight $\rho$, while its neighbouring slots share the remaining weight $\frac{1-\rho}{S-1}$. The updates are analogous to those in tile coding:

$$Q_{t+1}^{(T)}(O_t, A_t; k, i) \leftarrow Q_t^{(T)}(O_t, A_t; k, i) + \frac{\alpha}{T} \eta_{k,i} \delta_t, \tag{8}$$

where $\alpha$ is the learning rate of the transient updates, $T$ is the total number of tables, and $\delta_t$ is the TD-error at time $t$. The weighting factor $\eta_{k,i}$ reflects the contribution of slot $i$ in table $k$:

$$\eta_{k,i} = \begin{cases} \rho, & \text{if slot } i \text{ stores the query observation in the matched bin,} \\ \frac{1-\rho}{S-1}, & \text{if slot } i \text{ is a neighbour in the same bin,} \\ 0, & \text{otherwise.} \end{cases}$$

Although the transient table has a fixed size and older observations within a bin are evicted using a LRU strategy, it can still retain observations from a long time ago if temporally adjacent states are similar and therefore continue to map to the same bin, while observations encountered much later differ from those earlier states and are hashed into different bins, leaving the older ones untouched.

## 7 EXPERIMENTS: CRAFTAX-CLASSIC

We use the Craftax-Classic environment for large-scale experiments. The environment is open-ended, containing 22 achievements of varying difficulty. At each step $t$, the agent receives a $7 \times 9$ grid observation containing the object types in its view (Fig. 5), along with its inventory (e.g., wood, stone, or crafted tools) and intrinsic variables (e.g., health, hunger, or thirst). The agent can take one of 17 available actions. Within each episode, it receives a reward the first time it completes an achievement, such as collecting coal or crafting an iron pickaxe. Simpler achievements yield smaller rewards, while more advanced ones yield larger rewards. An episode ends when its length reaches 1000 steps, when the agent completes all 22 achievements, or when it is killed due to a zombie, skeleton, or by depletion of its intrinsic variables. More details can be found in Matthews et al. (2024).

We conducted two experiments: *online learning*[3], to demonstrate the effectiveness of our method in online CRL, and *benchmarking with* 250*M*, to show the competitiveness of our approach in a higher sample-complexity regime.

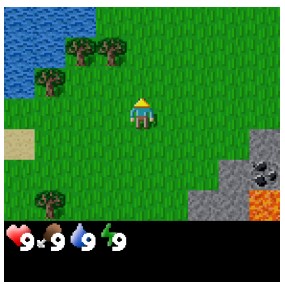

Figure 5: Craftax Env.

For the 250M benchmarking, the agent interacts with 1024 environments simultaneously, reducing the runtime for both our approach and the baselines to under 6 hours on a H100 GPU. *It is worth noting that we use parallel environments only to reduce wall-clock time and enable faster iteration. An ideal, general-purpose CRL algorithm should learn effectively from a single stream of experience.*

**Baselines:** We compare our approach against PQN, the state-of-the-art value-based algorithm on Craftax (Gallici et al., 2025), alongside three variants: PTQ-IHT, which utilizes Index Hash Tables (IHTs) rather than slot-based memory; TM-Only, which relies solely on the non-parametric transient approximator (Sec. 6); and PTQ-NeurIPS (Anand & Precup, 2023), a fully neural network-based baseline.

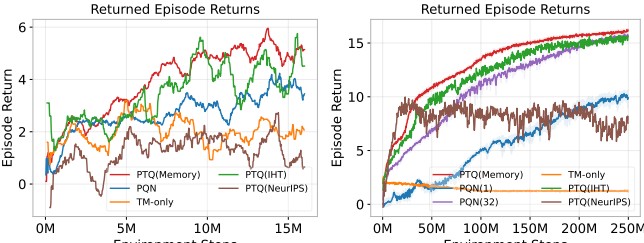

(a) Online Experiment Results      (b) 250M Experiment Results

Figure 6: PT(Memory) and PT(IHT) learns online, and outperforms other approaches on the 250M benchmark.

In this experiment, PQN aggregates data over 32 interaction steps. In the 250M benchmark, we evaluate PQN(1) (one gradient step per batch) and PQN(32) (32 gradient steps via minibatches), and in the online experiment, PQN uses 8 gradient steps. The hyperparameters for PQN were adopted from the PQN library.

For our approach, PTQ-Memory, we use a neural network to approximate the permanent value function and the non-parametric approximator described in Sec. 6 to approximate the transient value function. For the permanent neural network, we adopt the same hyperparameters and network architecture as PQN. For the transient memory, we tokenize the observation and append inventory, intrinsic values, and light level after quantizing (see Alg. 6 for details). The hash signature length is 256, which we split into two subsequences of 128 bits to obtain the bin indices for two tables. Each table has 2048 bins and 32 slots. We use a learning rate of 1.3/2 for each table, a decay factor of 0.95, and $\rho = 0.85$. For a fair comparison with baselines, we divide the data into 32 minibatches and update the permanent values toward the latest estimate of the overall value function, following Anand & Precup (2023). These hyperparameters were obtained by sweeping through candidate values for each parameter while keeping the others fixed. Additional details are provided in Appendix A.3. We used the same method to find the hyperparameters for baselines.

**Results:** The main results, averaged over 3 seeds for the online experiment (due to computational constraints) and 10 seeds for the 250M benchmarking, are shown in Fig. 6; per-achievement results are detailed in Appendix Fig. 10. Parameter ablation results are presented in the Appendix 12-17.

**Online Performance:** As shown in Fig. 6a, our approach, PTQ-Memory, successfully learns in the fully online setting. This success stems from the transient memory's ability to perform precise, local updates where generalization is explicitly controlled, preventing interference between states. In contrast, the PTQ-IHT ablation (mimicking Tile Coding) achieves fast initial updates but degrades over time. Due to the large observation space, the lack of slots and controlled generalization in IHTs leads to undesirable hash collisions. This results in over-generalization and poor long-term value estimation, empirically validating the necessity of our slot-based design. The PTQ-NeurIPS baseline, which relies on a neural network for transient estimates, fails completely in the online

---

[3]Strictly speaking it is a low-parallelism setting since the agent interacts with two instances of the environment (for 16M timesteps). We use two instances to reduce the experiment runtime to about 18 hours on a H100 GPU. We expect the results to carry over when the experience data source is reduced to a single stream.

setting. This confirms that gradient-based methods suffer from update instability and slow initial learning when denied large batches (Mnih et al., 2013; Elsayed et al., 2024). Similarly, the TM-only method performs poorly, confirming that transient memory alone lacks the capacity for effective long-term learning. Finally, PQN fails to learn rapidly in the low-parallelization setting as it relies on batching (32 steps) and environment parallelization to stabilize gradients.

**250M Benchmarking:** As shown in Fig. 6b and Fig. 11, our method outperforms all baselines in the extensive 250M step benchmark. The benefits are most evident in the early training stages: our approach surpasses a return of 10 in under 50M steps—twice as fast as PQN(32)—and reaches a return of 15 within 150M steps, whereas PQN(32) requires over 200M steps. It also achieves higher scores across all learnable achievements. While the PTQ-IHT variant performs competitively here, it still suffers marginally from collision-induced noise. This performance difference highlights the complementarity of our estimators: the transient component provides rapid, local feature discovery, which accelerates the permanent component's ability to generalize. Among PQN baselines, PQN(32) outperforms PQN(1) due to multiple gradient updates per step, though this minibatch strategy risks introducing primacy bias (Nikishin et al., 2022; D'Oro et al., 2022).

Overall, these results demonstrate that our approach is well-suited for online CRL on complex tasks.

## 8 EXPERIMENT: GENERALIZATION TO IMAGE-BASED TASKS

This section provides preliminary evidence of the generalizability of the non-parametric transient memory (Section 6) to image-based domains.

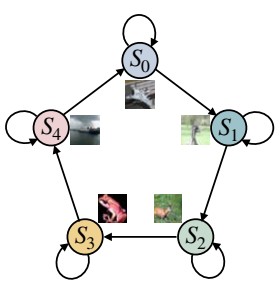

Figure 7: Image Task

**Task:** We evaluate our approach on a non-stationary, image-based MDP consisting of five states and five actions. Observations for each state are sampled from specific categories of the CIFAR-10 dataset. We evaluate three variations of this task by restricting the sub-sampled data size to 250, 500, and 1000 images per category. Transitions are stochastic: selecting the rewarding action (+1) leads to the next state w.p. $0.8$ (otherwise remaining current w.p. $0.2$), while suboptimal actions yield a small negative penalty and result in remaining in the current state w.p. $0.8$. Rewards are perturbed with Gaussian noise ($\mu = 0, \sigma = 0.01$). To induce non-stationarity, the optimal action is switched every $100k$ timesteps (from a set of three) over a $600k$ timestep duration. We use $\gamma = 0.9$. We compare the five algorithms from Section 7 (all fully online except PQN) and report mean rewards over 100 timesteps (90% C.I. over 30 seeds). Hyperparameter details are provided in Appendix A.4.

**Tokenization:** To enable MinHash hashing on image inputs, we first train a CIFAR-10 CNN classifier. We extract 256-dimensional features from the penultimate layer and binarize them via median thresholding. Our analysis revealed high similarity between these vectors (inter-class Hamming distance $\approx 130$ vs. intra-class $\approx 113$). We append positional indices (1 to 256) to the binary vector, analogous to Transformer positional encoding (Vaswani et al., 2017) and the spatial coordinates used in symbolic observations. This augmented vector is then processed by MinHash to map observations to transient memory slots.

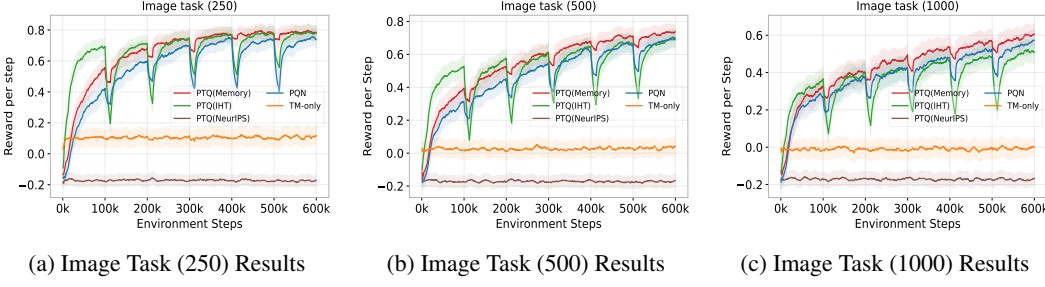

(a) Image Task (250) Results     (b) Image Task (500) Results     (c) Image Task (1000) Results

Figure 8: PTQ(Memory) variant adapts the fastest when changes occur in the environment.

**Results:** The results on Image MDPs, presented in Figure 8, mirror the trends observed in the Craftax experiments. The PTQ-Memory variant demonstrates superior performance. By utilizing memory slots to separate values and controlling generalization via $\rho$, it ensures precise transient updates. This mechanism enables rapid adaptation to task changes—a crucial advantage over the neural-network-based baselines. In contrast, PTQ-NeurIPS and TM-only fail to learn meaningful policies due to update instability and insufficient capacity, respectively. PQN achieves stability through batch updates but suffers from slow initial learning and slow recovery after task changes. Similarly, PTQ-IHT is hampered by the size of observation space, where increased hash collisions lead to detrimental over-generalization and degraded value estimates. These findings provide evidence that the memory-based Permanent-Transient (PT) approach successfully extends to complex visual domains.

## 9 DISCUSSION AND CONCLUSION

In this paper, we extended and enhanced the permanent-transient value function decomposition by incorporating separate feature representations to further improve performance through a better stability–plasticity trade-off. Specifically, we leveraged slowly evolving or static features—either hand-crafted (e.g., Fourier bases) or learned using neural networks—for the permanent component, alongside reward-predictive, fast-evolving, or non-parametric features for the transient component. This design yielded improved performance in both small-scale and large-scale experiments.

The backbone which allows our approach to scale is a novel MinHash-based non-parametric approximator that enables rapid online learning like tabular RL, local generalization like CMACs and tile coding (but controlled via a hyperparameter, $\rho$), and efficient storage and retrieval of observations and values, all while remaining modest in size relative to the complexity of the environment. We explore its use for estimating the transient value function in CRL, though its benefits may extend more broadly.

While tokenization is natural for symbolic observation, we leveraged a pre-trained convolutional neural network to obtain tokens in the image experiment. This setup allowed us to isolate and demonstrate the core contribution: the transient memory's ability to adapt instantly for a pixel-based observation. This requirement can be relaxed in future work by: leveraging a pretrained vision encoder or traditional CV techniques (bag of visual words) (Dosovitskiy, 2020; Radford et al., 2021); adapting deep hashing to bypass the tokenization step and directly compute a hash signature (Luo et al., 2023); exploiting the inductive biases of randomly initialized CNNs along with a small, trained projection layer to obtain the token vector (Farebrother et al., 2023); or simply treating individual pixels as tokens, analogous to the symbolic setting.

Our non-parameteric memory share some similarities with episodic memory (Pritzel et al., 2017): while both approaches use key-value storage, Episodic Memory typically acts as a non-parametric replay buffer that stores and retrieves specific past returns (or Q-value estimates) via complex kernel regression (averaging neighbours). In contrast, our non-parametric component is a function approximator. The values stored in our hash table are residuals, learned and updated via TD-error using simple summation. Consequently, our approach is designed for rapid adaptation in continual RL, rather than to accelerate single-task convergence.

Despite these contributions, many research questions remain open in the permanent–transient framework: extension to policy gradient algorithms; developing mechanisms for selective consolidation (determining when and what to transfer to permanent memory); integrating recurrent neural networks to fully realize memory; applying meta-learning to automate transient parameter tuning (Xu et al., 2018); and investigating other architectural choices for efficient permanent and transient learning. Additionally, combining our approach with neural networks-based continual learning strategies, such as EWC (Kirkpatrick et al., 2017), offers a promising direction to further stabilize long-term retention in the permanent component.

**Conclusion:** Ultimately, an RL agent's ability to continually learn from new experiences is crucial both for advancing our scientific understanding of intelligence and for building systems that perform reliably in real-world conditions. Our framework advances this goal by demonstrating that separate representations is critical for a better trade-off between stability and plasticity that scale to complex, non-stationary environments.

## REPRODUCIBILITY STATEMENT

We are committed to ensuring the reproducibility of our results. All code and configuration files required to reproduce our experiments will be released publicly upon acceptance. Our implementation builds on the publicly available purejaxql codebase, and we provide pseudocode for our approach in Appendix 6. Model architectures, hyperparameters, and training procedures are described in Section 7 and Appendix A.3. We use the publicly available Craftax environment (Matthews et al., 2024) for large-scale experiments, and all necessary details to reproduce the synthetic environments are included in the paper. Details on random seeds, hardware, and GPU usage are provided in the corresponding sections of the main paper. We used ChatGPT, Copilot, and Gemini for code auto-completion, beautifying plots, and developing hashing code in JAX. We also acknowledge the use of LLMs (ChatGPT, Apple writing tools, AI mode in Google Search, and Grammarly) for grammar correction and polishing certain parts of the paper.

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

## A APPENDIX

### A.1 THEORETICAL RESULTS

**Theorem 2** (Borkar, 1997). *Consider two $d$ and $l$ dimensional coupled iterates of the form:*

$$\theta_{t+1} \leftarrow \theta_t + \overline{\alpha}\big(\mathcal{A}(\theta_t, \mathbf{w}_t) + \mathcal{M}_{t+1}\big), \tag{9}$$

$$\mathbf{w}_{t+1} \leftarrow \mathbf{w}_t + \alpha\big(\mathcal{C}(\theta_t, \mathbf{w}_t) + \mathcal{N}_{t+1}\big), \tag{10}$$

*for $t \geq 0$. If,*

1. *$\mathcal{A} : \mathbb{R}^{d+l} \to \mathbb{R}^d$, $\mathcal{C} : \mathbb{R}^{d+l} \to \mathbb{R}^l$ are Lipschitz,*

2. *$\sum_t \overline{\alpha}_t = \sum_t \alpha_t = \infty$, $\sum_t \overline{\alpha}_t^2 = \sum_t \alpha_t^2 < \infty$, $\lim_{t \to \infty} \frac{\overline{\alpha}_t}{\alpha_t} \to 0$,*

3. *for $\mathcal{F}_t \triangleq \sigma(\theta_k, \mathbf{w}_k, \mathcal{M}_k, \mathcal{N}_k, k \leq t)$, $t \geq 0$, $(\mathcal{M}_t, \mathcal{F}_t)$, $(\mathcal{N}_t, \mathcal{F}_t)$ are sequences of random variables satisfying: $\sum_t \overline{\alpha}_t \mathcal{M}_t$, $\sum_t \alpha_t \mathcal{N}_t \leq \infty$ almost surely,*

*then the iterates converge converge almost surely to the asymptotically stable equilibria of the associated limiting ODEs.*

**Theorem 3** (Tsitsiklis & Van Roy (1996)). *Consider an iterative algorithm of the form $\mathbf{w}_{t+1} = \mathbf{w}_t + \alpha_t(\mathcal{A}(X_t)\mathbf{w}_t + b(X_t))$ where,*

1. *the step-size sequence $\alpha_t$ satisfies $\sum_t \alpha_t = \infty$, $\sum_t \alpha_t^2 < \infty$,*

2. *$X_t$ is a Markov process with a unique invariant distribution,*

3. *$\mathcal{A}(\cdot)$ and $b(\cdot)$ are matrix and vector valued functions respectively, for which $\mathbf{A} = \mathbb{E}_{d_\pi}[\mathcal{A}(X_t)]$ and $\mathbf{b} = \mathbb{E}_{d_\pi}[b(X_t)]$ are well defined and finite,*

4. *the matrix $\mathbf{A}$ is positive definite,*

5. *there exists constants $K$ and $q$ such that for all $X$*

   • *$\sum_{t=0}^\infty \|\mathbb{E}_\pi[\mathcal{A}(X_t)|X_0 = X] - \mathbf{A}\| \leq K(1 + h^q(X))$, and*
   • *$\sum_{t=0}^\infty \|\mathbb{E}_\pi[b(X_t)|X_0 = X] - \mathbf{b}\| \leq K(1 + h^q(X))$,*

6. *for any $q > 1$ there exists a constant $\mu_q$ such that for all $X, t$*

   • *$\mathbb{E}_\pi[h^q(X_t)|X_0 = X] \leq \mu_q(1 + h^q(X))$.*

*Then, $\mathbf{w}_t$ converges to $\mathbf{w}_\pi$, with probability one, where $\mathbf{w}_\pi$ is the unique vector that satisfies $\mathbf{A}\mathbf{w}_\pi + \mathbf{b} = 0$.*

**Lemma 1.** *For any fixed choice of $\theta$, for any task $\tau$ in the task distribution, the sequence of expected transient updates converges to a unique fixed point.*

*Proof.* We use the proof technique outlined in Anand & Precup (2021); Tsitsiklis & Van Roy (1996) to establish convergence.

**Step 1. Transient update as linear stochastic approximation.** Dropping the $(\tau, \pi)$ subscript for clarity, the transient update can be written as

$$\mathbf{w}_{t+1} = \mathbf{w}_t + \alpha_t\Big(R_{t+1} + \gamma(\theta^T \phi_{t+1} + \mathbf{w}_t^T z_{t+1}) - (\theta^T \phi_t + \mathbf{w}_t^T z_t)\Big)z_t$$

$$= \mathbf{w}_t + \alpha_t\big(b(X_t) + A(X_t)\mathbf{w}_t\big),$$

where

$$b(X_t) = z_t(R_{t+1} + \gamma\theta^T \phi_{t+1} - \theta^T \phi_t), \quad A(X_t) = z_t(z_t - \gamma z_{t+1})^T,$$

and $X_t = (S_t, S_{t+1}, \phi_t, z_t)$.

**Step 2. Limiting expectations.** Define
$$\mathbf{A} = \lim_{t \to \infty} \mathbb{E}[A(X_t)], \qquad \mathbf{b} = \lim_{t \to \infty} \mathbb{E}[b(X_t)].$$
Explicitly,
$$\mathbf{A} = \sum_s d(s)\, z(s) \Big( z(s) - \gamma \sum_{s'} [\mathcal{P}]_{ss'} z(s') \Big)^T = \boxed{Z^T D(I - \gamma \mathcal{P}) Z},$$
$$\mathbf{b} = \sum_s d(s)\, z(s) \Big( \mathcal{R}(s) + \gamma \theta^T \sum_{s'} [\mathcal{P}]_{ss'} \phi(s') - \theta^T \phi(s) \Big)$$
$$= \boxed{Z^T D\mathcal{R} + Z^T D(\gamma \mathcal{P} - I)\Phi\theta}.$$

$\mathbf{A}$ is positive definite because $D(I - \gamma \mathcal{P})$ has positive row sums (since $\gamma \mathcal{P}$ is a sub-stochastic matrix) and column sums (since $\mathbf{1}^T D(I - \gamma \mathcal{P}) = d^T - \gamma d^T = (1 - \gamma)d^T > 0$).

**Step 3. Bounded noise.** Using mixing of the Markov chain,
$$\|\mathbb{E}[A(X_t)|X_0] - \mathbf{A}\| = \|Z^T D_t(I - \gamma \mathcal{P})Z - Z^T D(I - \gamma \mathcal{P})Z\|$$
$$= \|Z^T(D_t - D)(I - \gamma \mathcal{P})Z\|$$
$$\leq \|Z^T\| \, \|D_t - D\| \, \|I - \gamma \mathcal{P}\| \, \|Z\|$$
$$\leq B_1 \cdot C\sigma^t \cdot B_2 \cdot B_1$$
$$\boxed{\leq K_1 \sigma^t},$$
$$\|\mathbb{E}[b(X_t)|X_0] - \mathbf{b}\| = \|Z^T(D_t - D)\mathcal{R} + Z^T(D_t - D)(\gamma \mathcal{P} - I)\Phi\theta\|$$
$$\leq \|Z^T\| \, \|D_t - D\| \, \|\mathcal{R}\| + \|Z^T\| \, \|D_t - D\| \, \|(\gamma \mathcal{P} - I)\| \, \|\Phi\theta\|$$
$$\boxed{\leq K_2 \sigma^t}.$$
Therefore,
$$\sum_{t=0}^{\infty} \|\mathbb{E}[A(X_t)|X_0] - \mathbf{A}\| \leq \sum_{t=0}^{\infty} K_1 \sigma^t = \boxed{\frac{K_1}{1 - \sigma} = \bar{K}_1},$$
$$\sum_{t=0}^{\infty} \|\mathbb{E}[b(X_t)|X_0] - \mathbf{b}\| \leq \sum_{t=0}^{\infty} K_2 \sigma^t = \boxed{\frac{K_2}{1 - \sigma} = \bar{K}_2}.$$

**Step 4. Fixed point.** Thus, the expected iterates converge to the unique fixed point
$$\mathbf{w}_\tau^* = \mathbf{A}^{-1}\mathbf{b}$$
$$= Z^T D(I - \gamma \mathcal{P})Z)^{-1}(Z^T D\mathcal{R} + Z^T D(\gamma \mathcal{P} - I)\Phi\theta)$$
$$= Z^T D(I - \gamma \mathcal{P})Z)^{-1} Z^T D\mathcal{R} + Z^T D(I - \gamma \mathcal{P})Z)^{-1}(Z^T D(\gamma \mathcal{P} - I)V_\theta^{(P)})$$
$$= \boxed{\mathbf{w}_{Z,\tau}^{(TD)} - \big(Z^T D_\tau(I - \gamma \mathcal{P}_{\pi,\tau})Z\big)^{-1} Z^T D_\tau(I - \gamma \mathcal{P}_{\pi,\tau})\Phi\theta^*}$$

$\square$

**Lemma 2.** *The sequence of expected permanent updates converges to a unique fixed point.*

*Proof.* **Step 1. Permanent update with transient fixed point.** Since permanent updates evolve on a slower timescale, we may treat the transient parameters as converged. The update is
$$\theta_{t+1} = \theta_t + \overline{\alpha}_t\, C(X_t), \quad \text{where } C(X_t) = \mathbf{w}_t^T z_t \phi_t.$$
Define
$$\mathbf{C} = \sum_\tau p(\tau) \sum_s d_\tau(s) \mathbf{w}_\tau^T z(s) \phi(s)$$
$$= \boxed{\sum_\tau p(\tau) \Phi^T D_\tau Z \mathbf{w}_\tau^*}$$

**Step 2. Boundedness and Lipschitz condition.** For each task $\tau$,

$$\mathbf{C}_\tau = \Phi^T D_\tau Z \mathbf{w}_\tau^*,$$

and using $\mathbf{w}_\tau^* = \mathbf{A}_\tau^{-1}\big(Z^T D_\tau \mathcal{R}_\tau + Z^T D_\tau(\gamma \mathcal{P}_\tau - I)\Phi\theta\big)$, we obtain

$$
\begin{aligned}
\|\mathbf{C}_\tau\| &= \big\|\Phi^T D_\tau Z \mathbf{w}_\tau^*\big\| \\
&= \big\|\Phi^T D_\tau Z \mathbf{A}_\tau^{-1}(Z^T D_\tau \mathcal{R}_\tau + Z^T D_\tau(\gamma \mathcal{P}_\tau - I)\Phi\theta)\big\| \\
&\leq \big\|\Phi^T D_\tau Z \mathbf{A}_\tau^{-1} Z^T D_\tau\big\| + \big\|(\mathcal{R}_\tau + (\gamma\mathcal{P}_\tau - I)\Phi\theta)\big\| \\
&\leq K_3, \\
\|\mathbf{C}\| &= \bigg\|\sum_\tau p(\tau)\Phi^T D_\tau Z \mathbf{w}_\tau\bigg\| \\
&\leq \sum_\tau \big\|\Phi^T D_\tau Z \mathbf{w}_\tau\big\| \\
&\leq K_3.
\end{aligned}
$$

Thus the mapping is bounded and Lipschitz.

**Step 3. Noise boundedness.** Because tasks are sampled i.i.d. and each task's Markov chain is rapidly mixing, the noise terms have finite variance.

**Step 4. Fixed point.** Therefore, the expected permanent updates converge to the unique fixed point:

$$
\begin{aligned}
\mathbf{C} &= \sum_\tau p(\tau)\Phi^T D_\tau Z \mathbf{w}_\tau^* \\
0 &= \sum_\tau p(\tau)\Big(\Phi^T D_\tau Z v_{Z,\tau}^{(TD)} - \Phi^T D_\tau Z \big(Z^T D_\tau(I - \gamma\mathcal{P}_{\pi,\tau})Z\big)^{-1} Z^T D_\tau(I - \gamma\mathcal{P}_{\pi,\tau})\Phi\theta^*\Big) \\
0 &= \mathbb{E}_\tau[\Phi^T D_\tau Z v_{Z,\tau}^{(TD)}] - \mathbb{E}_\tau[\Phi^T D_\tau \Pi_{Z,\tau}\Phi\theta^*]
\end{aligned}
$$

$$\mathbb{E}_\tau[\Phi^T D_\tau \Pi_{Z,\tau}\Phi]\theta^* = \mathbb{E}_\tau[\Phi^T D_\tau Z v_{Z,\tau}^{(TD)}]$$

$$\boxed{\theta^* = \mathbb{E}_\tau[\Phi^T D_\tau \Pi_{Z,\tau}\Phi]^{-1}\mathbb{E}_\tau[\Phi^T D_\tau Z v_{Z,\tau}^{(TD)}]}$$

$\square$

---

**Corollary 2.** *If* $Z = \Phi$, *then*

$$\theta^* = \mathbb{E}_\tau\big[\Phi^T D_\tau \Phi\big]^{-1}\mathbb{E}_\tau\big[\Phi^T D_\tau \Phi \mathbf{w}_{\Phi,\tau}^{(TD)}\big],$$

$$\mathbf{w}_\tau^* = \mathbf{w}_{\Phi,\tau}^{(TD)} - \theta^*.$$

*Moreover, in the single-task setting,* $\theta^* = \mathbf{w}_{\Phi,\tau}^{(TD)}$ *and* $\mathbf{w}_\tau^* = 0$.

---

*Proof.* When $\Phi = Z$,

$$\Pi_{Z,\tau} = \Pi_{\Phi,\tau} = \Phi\big(\Phi^T D_\tau(I - \gamma\mathcal{P}_{\pi,\tau})\Phi\big)^{-1}\Phi^T D_\tau(I - \gamma\mathcal{P}_{\pi,\tau}),$$

$$\Pi_{\Phi,\tau}\Phi = \Phi\big(\Phi^T D_\tau(I - \gamma\mathcal{P}_{\pi,\tau})\Phi\big)^{-1}\big(\Phi^T D_\tau(I - \gamma\mathcal{P}_{\pi,\tau})\Phi\big) = \Phi.$$

Therefore,

$$
\begin{aligned}
\theta^* &= \mathbb{E}_\tau[\Phi^T D_\tau \Phi]^{-1}\mathbb{E}_\tau[\Phi^T D_\tau \Phi \mathbf{w}_\Phi^{(TD)}] \\
\mathbf{w}_\tau^* &= \mathbf{w}_{\Phi,\tau}^{(TD)} - \big(\Phi^T D_\tau(I - \gamma\mathcal{P}_{\pi,\tau})\Phi\big)^{-1}\Phi^T D_\tau(I - \gamma\mathcal{P}_{\pi,\tau})\Phi\theta^*, \\
&= \mathbf{w}_{\Phi,\tau}^{(TD)} - \theta^*.
\end{aligned}
$$

In the single task setting,

$$\theta^* = \mathbb{E}_\tau[\Phi^T D_\tau \Phi]^{-1} \mathbb{E}_\tau[\Phi^T D_\tau \Phi \mathbf{w}_\Phi^{(TD)}],$$
$$= \left(\Phi^T D_\tau \Phi\right)^{-1} \left(\Phi^T D_\tau \Phi\right) \mathbf{w}_{\Phi,\tau}^{(TD)} = \mathbf{w}_{\Phi,\tau}^{(TD)},$$
$$\mathbf{w}_\tau^* = \mathbf{w}_{\Phi,\tau}^{(TD)} - \theta^* = \mathbf{w}_{\Phi,\tau}^{(TD)} - \mathbf{w}_{\Phi,\tau}^{(TD)} = 0.$$

$\square$

## A.2 SMALL-SCALE EXPERIMENTS

**Pseudocode for prediction and control with separate permanent and transient features.**

---
**Algorithm 1** Prediction with Linear Approximations
---
1: **Initialize:** buffer $\mathcal{B}$, parameters $\theta$, $\mathbf{w}$
2: **for** $t = 0 \to \infty$ **do**
3:    Take action $A_t$
4:    Store state $S_t$ in $\mathcal{B}$
5:    Observe reward $R_{t+1}$ and next state $S_{t+1}$
     # Update transient parameters
6:    $\mathbf{w}_{t+1} \leftarrow \mathbf{w}_t + \alpha\big(R_{t+1} + \gamma V^{(PT)}(S_{t+1}) - V^{(PT)}(S_t)\big)z(S_t)$
7:    **if** Task ends **then**
8:       **for** Every $S_k$ in $\mathcal{B}$ **do**
        # Update permanent parameters
9:          $\theta_{k+1} \leftarrow \theta_k + \overline{\alpha}\left(V^{(PT)}(S_k) - V^{(P)}(S_k)\right)\phi(S_k)$
10:      **end for**
       # Reset transient parameters
11:        $\mathbf{w}_{t+1} \leftarrow 0$
       # Clear buffer
12:        Reset $\mathcal{B}$
13:    **end if**
14: **end for**
---

---
**Algorithm 2** Control with Linear Approximations
---
1: **Initialize:** buffer $\mathcal{B}$, parameters $\theta$, $\mathbf{w}$
2: **for** $t = 0 \to \infty$ **do**
3:    Take action $A_t$
4:    Store state $S_t$, $A_t$ in $\mathcal{B}$
5:    Observe reward $R_{t+1}$ and next state $S_{t+1}$
     # Update transient parameters
6:    $\mathbf{w}_{t+1} \leftarrow \mathbf{w}_t + \alpha\big(R_{t+1} + \gamma \max_{a'} Q^{(PT)}(S_{t+1}, a') - Q^{(PT)}(S_t, A_t)\big)z(S_t, A_t)$
7:    **if** Task ends **then**
8:       **for** Every $(S_k, A_k)$ in $\mathcal{B}$ **do**
        # Update permanent parameters
9:          $\theta_{k+1} \leftarrow \theta_k + \overline{\alpha}\left(Q^{(PT)}(S_k, A_k) - Q^{(P)}(S_k, A_k)\right)\phi(S_k, A_k)$
10:      **end for**
       # Reset transient parameters
11:        $\mathbf{w}_{t+1} \leftarrow 0$
       # Clear buffer
12:        Reset $\mathcal{B}$
13:    **end if**
14: **end for**
---

**Hyperparameter Sweeps for Linear Prediction (best highlighted in bold).**

```
TD-learning:
    LR = [3e-2, 1e-2, 3e-3, 1e-3, 3e-4, 1e-4]

TD-learning (Reset):
    LR = [3e-2, 1e-2, 3e-3, 1e-3, 3e-4]

PT-TD (NeurIPS):
    LR-P = [1e-2, 3e-3, 1e-3, 3e-4, 1e-4, 3e-5]
    LR-T = [1e-1, 3e-2, 1e-2, 3e-3, 1e-3, 3e-4]

PT-TD (Ours):
    LR-P = [1e-2, 3e-3, 1e-3, 3e-4, 1e-4, 3e-5]
    LR-T = [1e-1, 3e-2, 1e-2, 3e-3, 1e-3, 3e-4]
```

**Hyperparameter Sweeps for Linear Control (best highlighted in bold).**

```
Q-learning:
    LR = [0.5, 0.3, 0.1, 0.03, 0.01, 0.003, 0.001]

Q-learning (Reset):
    LR = [0.5, 0.3, 0.1, 0.03, 0.01, 0.003, 0.001]

PT-Q (NeurIPS):
    LR-P = [0.03, 0.01, 0.003, 0.001, 0.0003]
    LR-T = [0.5, 0.3, 0.1, 0.03, 0.01]

PT-Q (Ours):
    LR-P = [0.03, 0.01, 0.003, 0.001, 0.0003]
    LR-T = [0.5, 0.3, 0.1, 0.03, 0.01]
```

**Tasks Used in Experiments**

| Task | G1 | G2 | G3 | G4 |
|------|----|----|----|----|
| **1** | 0 | 1 | 0 | 1 |
| **2** | 1 | 0 | 1 | 0 |
| **3** | 0 | 0 | 1 | 1 |
| **4** | 1 | 1 | 0 | 0 |

Table 1: Tasks used in Linear Prediction Experiments.

| Task | 🔴 | 🔵 |
|------|------|------|
| **1** | 1 | 0.1 |
| **2** | 0.1 | 1 |

Table 2: Tasks Used in Linear Control Experiments

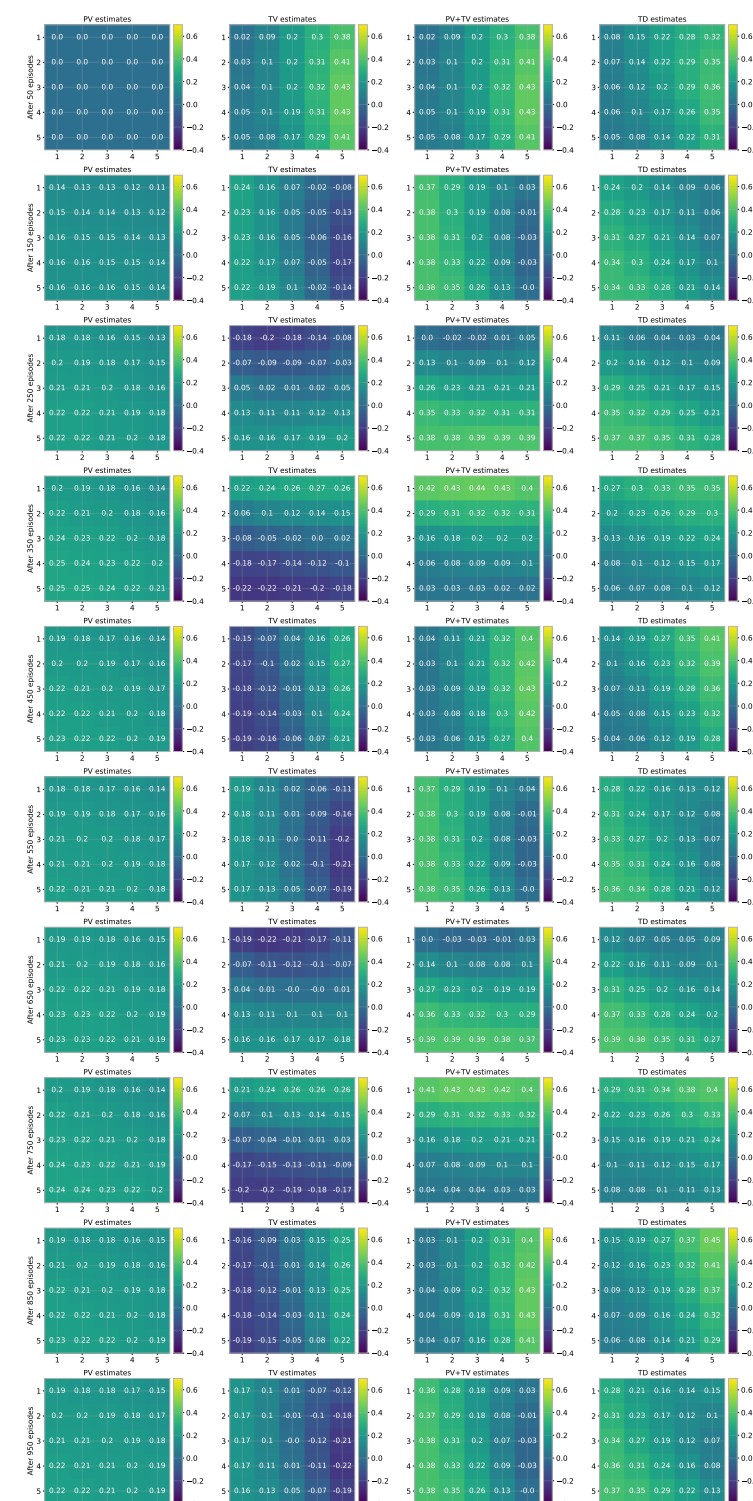

Figure 9: Value Function Heatmap for the Prediction Task

## A.3 CRAFTAX EXPERIMENTS

**Hyperparameter Tuning.**

Craftax Baselines: For the baselines and the permanent network in all variants of PTQ-learning, we used hyperparameters from the PQN repository, which are consistent with the results published in the PQN paper. We only tuned minibatches for the PQN baseline, because it determined the number of steps the permanent network would take, which we also tuned for our approach. Transient parameters were found by performing a search over a range of values. Craftax: Given the large hyperparameter space, we used coordinate-wise search: we fixed all hyperparameters but one to a reasonable baseline and performed a grid search for that single hyperparameter. This process was then repeated iteratively for each hyperparameter, fixing the newly tuned value before moving to the next. For each configuration, we ran a total of 150M steps with 1024 environments in parallel, using AUC as the selection metric. The final results were reported for 250M steps. We used the same HPs for the online craftax experiment (minibatch was reduced to 8 to fit the smaller batch size).

Transient in Craftax: Given the large hyperparameter space, we used coordinate-wise search: we fixed all hyperparameters but one to a reasonable baseline and performed a grid search for that single hyperparameter. This process was then repeated iteratively for each hyperparameter, fixing the newly tuned value before moving to the next. For each configuration, we ran a total of 150M steps with 1024 environments in parallel, using AUC as the selection metric. The final results were reported for 250M steps. We used the same HPs for the online craftax experiment (minibatch was reduced to 8 to fit the smaller batch size).

**Role of HPs.**

1. **Tables ($T$) vs. Generalization:** Similar to tiles in Tile Coding. More tables spread information, increasing generalization. Fewer tables concentrate information, reducing generalization.

2. **Slots ($S$) & Eviction:** Determines capacity. Too few slots lead to high eviction rates; too many slots approach tabular memory.

3. **Generalization ($\rho$):** Explicitly controls the mixing of values between slots. $\rho = 1$ forces isolation (no generalization); $\rho < 1$ enables local smoothing.

4. **Number of Hashes ($K$):** Determines the precision of the MinHash signature. A higher $K$ better preserves the Jaccard similarity property (improving retrieval accuracy), though with diminishing returns beyond a certain point.

**Non-parametric Transient Memory.** The non-parametric transient memory is implemented through three core methods: *Put*, which stores incoming observations; *Get*, which retrieves stored value estimates; and *UpdateTDError*, which updates the estimates in proportion to their contribution. Pseudocode for these methods is given in Algorithms 3,4, and 5.

---

**Algorithm 3** PUT method for non-parametric transient memory

---

1: **procedure** PUT($\mathcal{M}, x$)
**Require:** $\mathcal{M}$: transient memory; $x$: observation
**Ensure:** Updated memory $\mathcal{M}$
2:     $(sig, tag) \leftarrow$ MINHASH($x$)
3:     $bins \leftarrow$ GETBUCKETS($sig$)
      # Insert observation in all tables
4:     **for** $t \leftarrow 0$ **to** $T - 1$ **do**
5:         **if** (CONTAINSTAG($\mathcal{M}[t, bins[t]], tag$) == **false**) **then**
          # empty-first else LRU
6:             $slot \leftarrow$ SELECTSLOT($\mathcal{M}, t, bins[t]$)
7:             $\mathcal{M}[t, bins[t], slot].tag \leftarrow tag$
          # mean over other valid slots
8:             $\mathcal{M}[t, bins[t], slot].values \leftarrow$ INITIALIZE($t, bins[t]$)
9:         **else**
10:            $slot \leftarrow$ FINDSLOTBYTAG($\mathcal{M}[t, bins[t]], tag$)
11:        **end if**
12:        $\mathcal{M}[t, bins[t], slot].age \leftarrow$ TOUCH($\mathcal{M}.clock$)
13:    **end for**
14:    $\mathcal{M}.clock \leftarrow \mathcal{M}.clock + 1$
15:    **return** $\mathcal{M}$
16: **end procedure**

---

**Algorithm 4** GET method for non-parametric transient memory

---

1: **procedure** GET($\mathcal{M}, x, \rho$)
**Require:** $\mathcal{M}$: transient memory; $x$: observation; $\rho$: mixing weight
**Ensure:** values
2:     $(sig, tag) \leftarrow$ MINHASH($x$)
3:     bins $\leftarrow$ GETBUCKETS(sig)
4:     values $\leftarrow 0$
      # Collect values from every table
5:     **for** $t \leftarrow 0$ **to** $T - 1$ **do**
6:         slot $\leftarrow$ FINDSLOTBYTAG($\mathcal{M}[t, bins[t]], tag$)
      # Weigh the update for matching slot by $\rho$
7:         values $\leftarrow$ values $+\rho \cdot \mathcal{M}[t, bins[t], slot].values$
8:         OSlots $\leftarrow$ FINDOTHERVALIDSLOTS($\mathcal{M}[t, bins[t]], tag$)
9:         $n \leftarrow$ LEN(OSlots)
      # Divide weight $(1 - \rho)$ equally among other valid slots
10:        **for** $s \leftarrow 0$ **to** $n - 1$ **do**
11:            values $\leftarrow$ values $+\frac{(1-\rho)}{n} \cdot \mathcal{M}[t, bins[t], OSlots[s]].values$
12:        **end for**
13:    **end for**
14:    **return** values
15: **end procedure**

---

A.4   IMAGE EXPERIMENTS

We performed a grid search over all the hyperparameters for the 256-image variant. Once we found the best values, we fixed them and tested the performance for the 500- and 1000-image variants (see Figures 18-20. The best hyperparameter values for each algorithm is provided below:

---

**Algorithm 5** UpdateTDError method for non-parametric transient memory

---

1: **procedure** UPDATETDERROR($\mathcal{M}, x, a, \rho, \delta, \alpha$)
**Require:** $\mathcal{M}$: transient memory; $x$: observation; $a$: action; $\rho$: mixing weight; $\delta$: TD-error; $\alpha$: learning rate
**Ensure:** Updated memory $\mathcal{M}$
2:    $(sig, tag) \leftarrow$ MINHASH($x$)
3:    bins $\leftarrow$ GETBUCKETS(sig)
4:    **for** $t \leftarrow 0$ **to** $T - 1$ **do**
5:       slot $\leftarrow$ FINDSLOTBYTAG($\mathcal{M}[t, bins[t]], tag$)
       # Weight the matching slot by $\rho$
6:       $\mathcal{M}[t, bins[t], slot].values[a] \leftarrow \mathcal{M}[t, bins[t], slot].values[a] + \frac{\alpha}{T}\rho\delta$
7:       OSlots $\leftarrow$ FINDOTHERVALIDSLOTS($\mathcal{M}[t, bins[t]], tag$)
8:       $n \leftarrow$ LEN($OSlots$)
       # Divide TD-error equally among other valid slots by weighting by $(1 - \rho)$
9:       **for** $s \leftarrow 0$ **to** $n - 1$ **do**
10:          $\mathcal{M}[t, bins[t], OSlots[s]].values[a] \leftarrow \mathcal{M}[t, bins[t], OSlots[s]].values[a] + \frac{\alpha}{nT}(1-\rho)\delta$
11:       **end for**
12:    **end for**
13:    **return** $\mathcal{M}$
14: **end procedure**

---

| Parameter | Value |
|---|---|
| ALG_NAME | PQN |
| TOTAL_TIMESTEPS | $5 \times 10^8$ |
| TOTAL_TIMESTEPS_DECAY | $5 \times 10^8$ |
| NUM_ENVS | 1024 |
| NUM_STEPS | 32 |
| EPS_START | 0.1 |
| EPS_FINISH | 0.005 |
| EPS_DECAY | 0.2 |
| NUM_MINIBATCHES | 1 for PQN(1), 32 for PQN(32) |
| NUM_EPOCHS | 1 |
| NORM_INPUT | True |
| NORM_TYPE | layer_norm |
| HIDDEN_SIZE | 1024 |
| NUM_LAYERS | 4 |
| LR | 0.0001 |
| MAX_GRAD_NORM | 1.0 |
| LR_LINEAR_DECAY | True |
| REW_SCALE | 1.0 |
| GAMMA | 0.99 |
| Q_LAMBDA | False |
| LAMBDA | 0 |
| **Environment** | |
| ENV_NAME | Craftax-Classic-Symbolic-v1 |
| USE_OPTIMISTIC_RESETS | True |
| OPTIMISTIC_RESET_RATIO | 16 |
| LOG_ACHIEVEMENTS | True |

Table 3: Hyperparameters for PQN.

**Algorithm 6** PTQ for Craftax

**Require:** $\mathcal{M}$: transient memory; $\theta$: permanent parameters; env: environment; $T_s$: total timesteps; $\mathcal{B}$: buffer; $\overline{\alpha}$: permanent LR; $\alpha$: transient LR; $\rho$: mixing weight; $\epsilon$: exploration rate; $\gamma$: discount; $k$: PM update period; $\lambda$: TM decay
**Ensure:** Updated $\theta, \mathcal{M}$
1: $s \leftarrow$ env.reset()
2: $\mathcal{M} \leftarrow \text{PUT}(\mathcal{M}, s)$
3: **for** $t \leftarrow 1$ **to** $T_s$ **do**
   # permanent values
4:    $Q^{(P)}(s) \leftarrow \text{GETPERMANENT}(s, \theta)$
   # transient values
5:    $Q^{(T)}(s) \leftarrow \text{GET}(\mathcal{M}, s, \rho)$
   # Compute $Q$ for current state
6:    $Q^{(PT)}(s) \leftarrow Q^{(P)}(s) + Q^{(T)}(s)$
   # Select action (epsilon-greedy over $Q$)
7:    $a \leftarrow \text{EPSILONGREEDY}(Q^{(PT)}(s), \epsilon)$
   # Step environment
8:    $(s', r) \leftarrow$ env.step($a$)
9:    $\mathcal{M} \leftarrow \text{PUT}(\mathcal{M}, s')$
10:    $\mathcal{B} \leftarrow \mathcal{B} \cup \{(s, a, Q^{(P)}(s))\}$
   # Evaluate $Q$ for next state
11:    $Q^{(P)}(s') \leftarrow \text{GETPERMANENT}(s', \theta)$
12:    $Q^{(T)}(s') \leftarrow \text{GET}(\mathcal{M}, s', \rho)$
13:    $Q^{(PT)}(s') \leftarrow Q^{(P)}(s') + Q^{(T)}(s')$
   # TD error and transient update
14:    $\delta \leftarrow r + \gamma \max_{a'} Q^{(PT)}(s', a') - Q^{(PT)}(s, a)$
15:    $\mathcal{M} \leftarrow \text{UPDATETDERROR}(\mathcal{M}, s, a, \rho, \delta, \alpha)$
   # Periodic permanent update and TM decay
16:    **if** $\text{mod}(t, k) = 0$ **then**
17:       $\theta \leftarrow \text{UPDATEPM}(\mathcal{M}, \mathcal{B}, \overline{\alpha})$
18:       $\mathcal{B} \leftarrow \{\}$
19:       $\mathcal{M} \leftarrow \text{DECAYVALUES}(\mathcal{M}, \lambda)$
20:    **end if**
21:    $s \leftarrow s'$
22: **end for**
23: **return** $\theta, \mathcal{M}$

| Parameter | Value |
|---|---|
| **Permanent Memory** | |
| ALG_NAME | PTQ |
| TOTAL_TIMESTEPS | $2 \times 10^7$ |
| TOTAL_TIMESTEPS_DECAY | $5 \times 10^8$ |
| NUM_ENVS | 1024 |
| NUM_STEPS | 32 (Permanent memory update frequency) |
| EPS_START | 0.1 |
| EPS_FINISH | 0.005 |
| EPS_DECAY | 0.2 |
| NUM_MINIBATCHES | 32 |
| NUM_EPOCHS | 1 |
| NORM_INPUT | True |
| NORM_TYPE | layer_norm |
| HIDDEN_SIZE | 1024 |
| NUM_LAYERS | 4 |
| LR | 0.0001 |
| MAX_GRAD_NORM | 1.0 |
| LR_LINEAR_DECAY | True |
| REW_SCALE | 1.0 |
| GAMMA | 0.99 |
| Q_LAMBDA | False |
| LAMBDA | 0 |
| **Environment** | |
| ENV_NAME | Craftax-Classic-Symbolic-v1 |
| USE_OPTIMISTIC_RESETS | True |
| OPTIMISTIC_RESET_RATIO | 16 |
| LOG_ACHIEVEMENTS | True |
| **Transient Memory** | |
| NUM_TABLES | 2 |
| TRANSIENT_TABLE_SIZE | 2048 |
| NUM_SLOTS | 32 |
| NUM_HASHES | 128 |
| CROP_SIZE | 7 |
| $\rho$ | 0.85 |
| TRANSIENT_LR | 1.3 |
| DECAY | 0.95 |

Table 4: Hyperparameters for PTQ for the results presented in the main paper.

| Parameter | Value |
|---|---|
| **Hyperparameters** | |
| ALG_NAME | PQN |
| NUM_MINIBATCHES | 4 |
| NUM_EPOCHS | 1 |
| NORM_INPUT | True |
| NORM_TYPE | layer_norm |
| LR | 0.001 |
| MAX_GRAD_NORM | 1.0 |

Table 5: Hyperparameters for PQN for Image task.

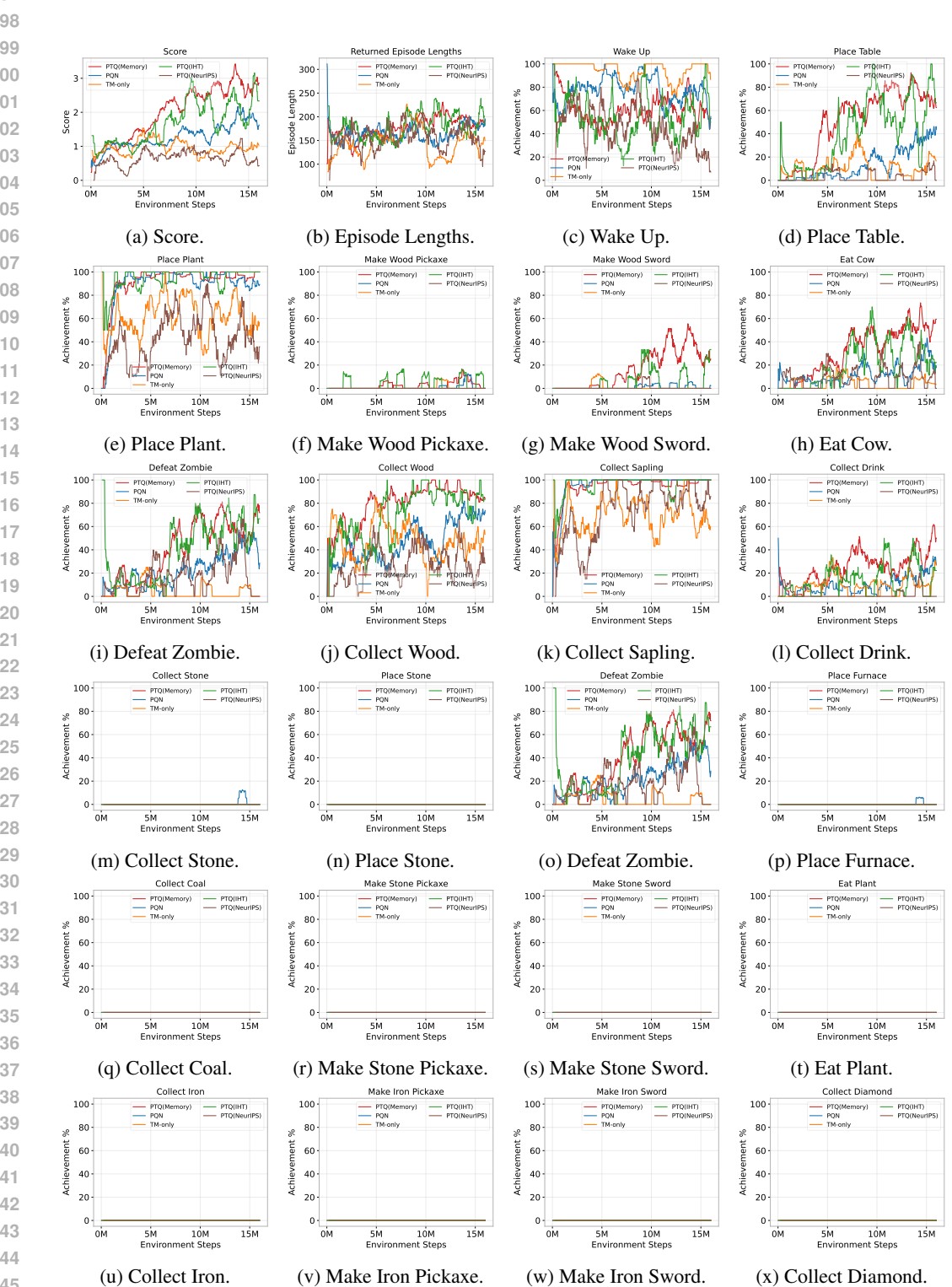

Figure 10: All achievements in the craftax online experiment.

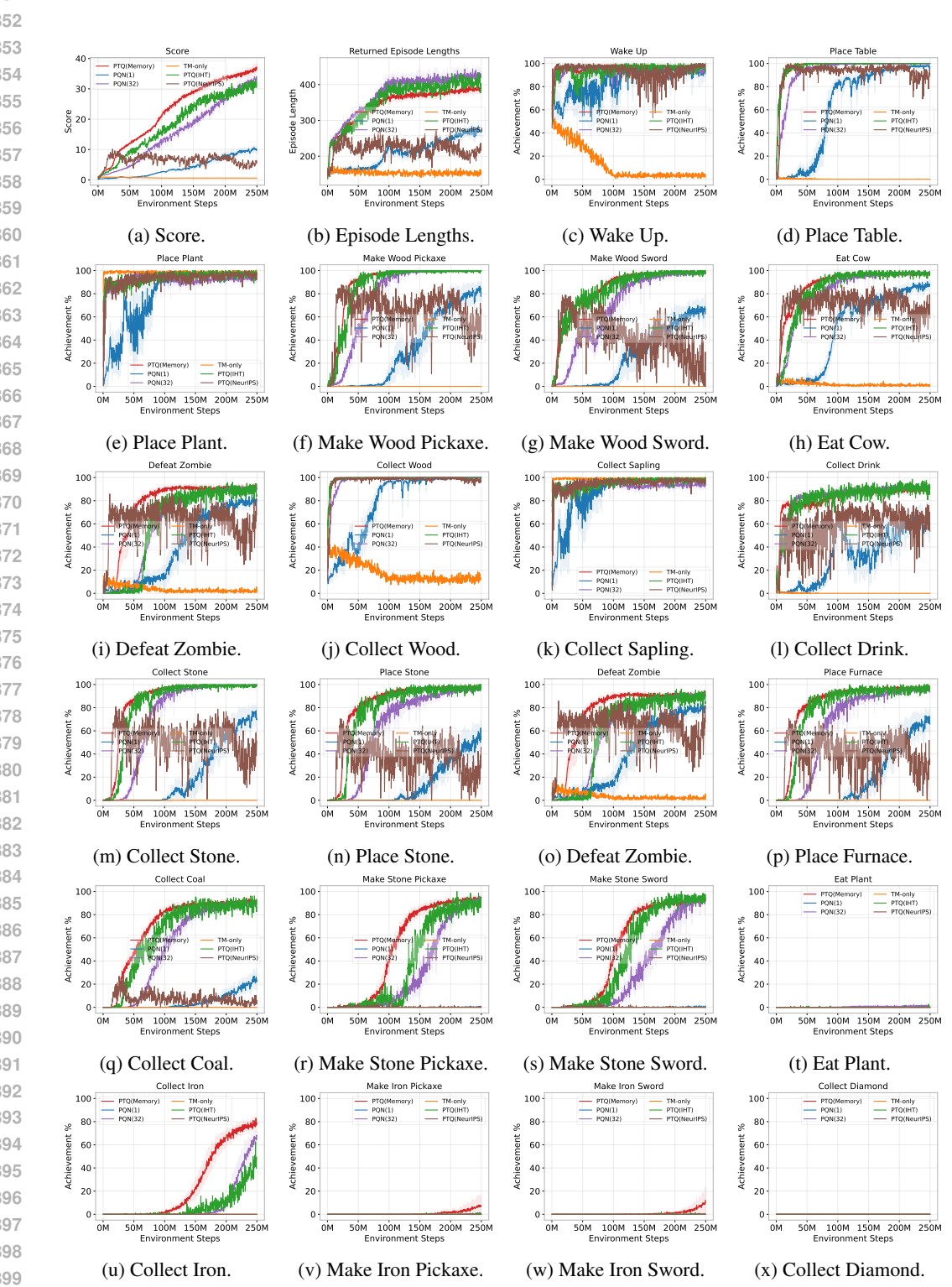

(a) Score.     (b) Episode Lengths.     (c) Wake Up.     (d) Place Table.

(e) Place Plant.     (f) Make Wood Pickaxe.     (g) Make Wood Sword.     (h) Eat Cow.

(i) Defeat Zombie.     (j) Collect Wood.     (k) Collect Sapling.     (l) Collect Drink.

(m) Collect Stone.     (n) Place Stone.     (o) Defeat Zombie.     (p) Place Furnace.

(q) Collect Coal.     (r) Make Stone Pickaxe.     (s) Make Stone Sword.     (t) Eat Plant.

(u) Collect Iron.     (v) Make Iron Pickaxe.     (w) Make Iron Sword.     (x) Collect Diamond.

Figure 11: All achievements in the craftax 250M experiment.

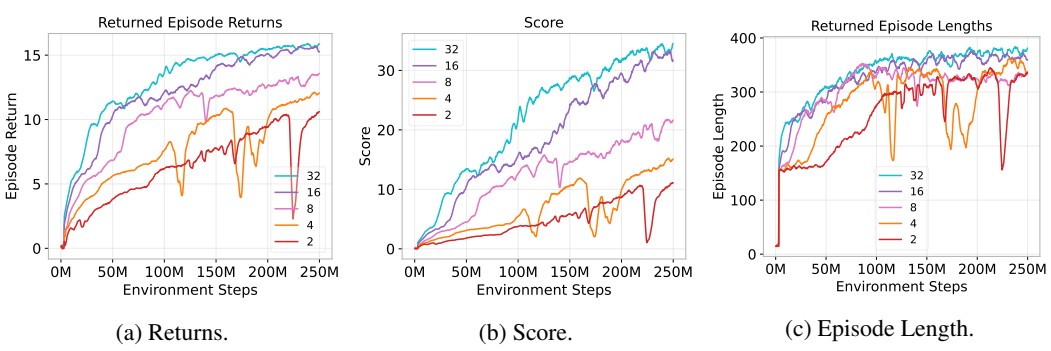

(a) Returns.

(b) Score.

(c) Episode Length.

Figure 12: Effect of minibatch updates to permanent network on performance.

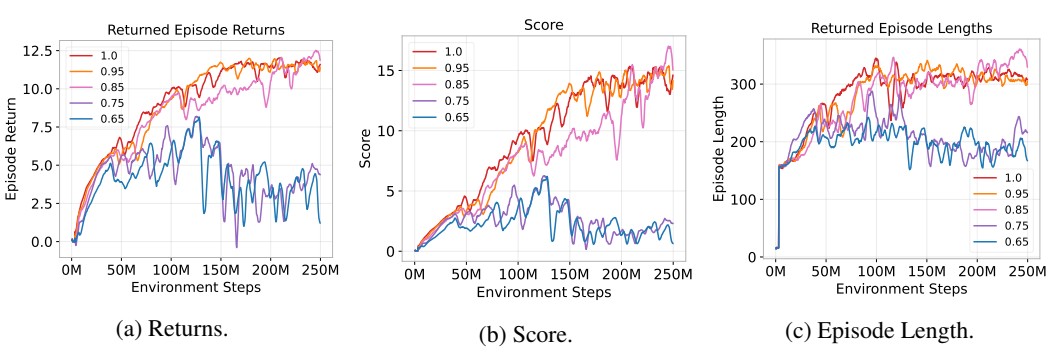

(a) Returns.

(b) Score.

(c) Episode Length.

Figure 13: Effect of $\rho$ on performance.

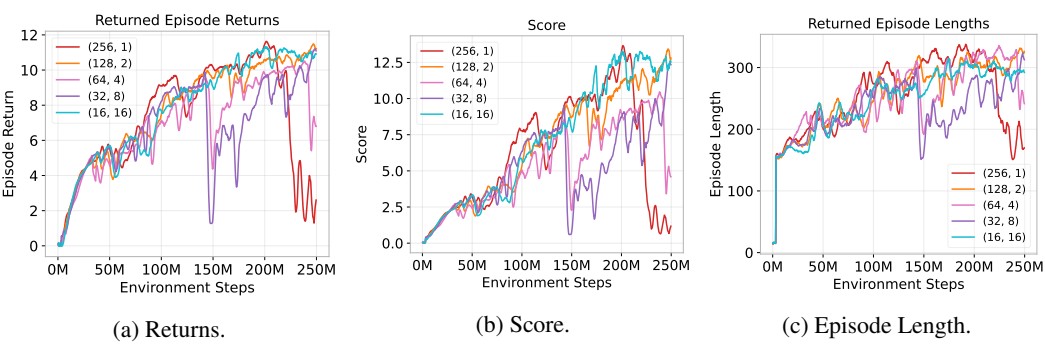

(a) Returns.

(b) Score.

(c) Episode Length.

Figure 14: Effect of number of tables on Performance (fixed hash signature to 256 bits).

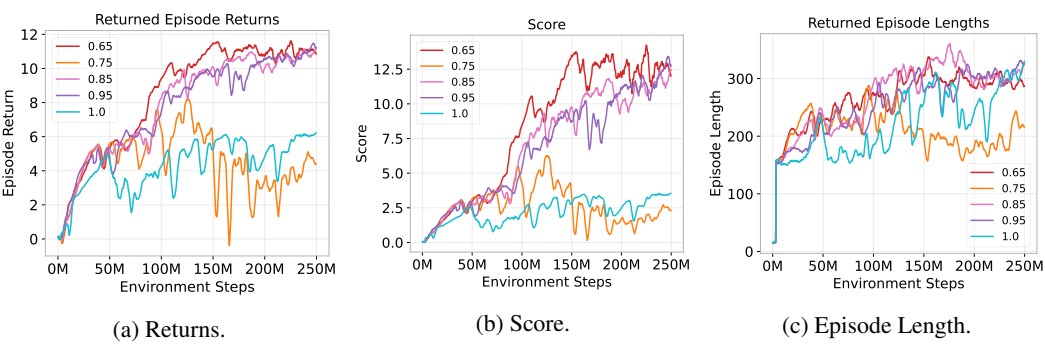

(a) Returns.

(b) Score.

(c) Episode Length.

Figure 15: Effect of the decay parameter on performance.

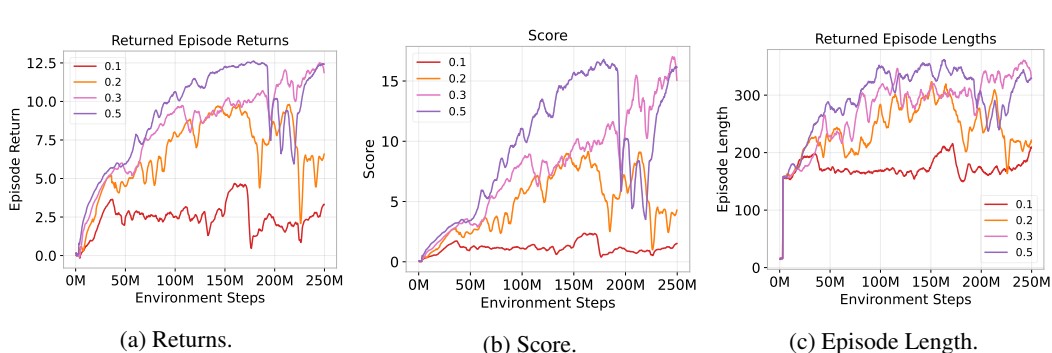

(a) Returns.
(b) Score.
(c) Episode Length.

Figure 16: Effect of the transient learning rate on performance.

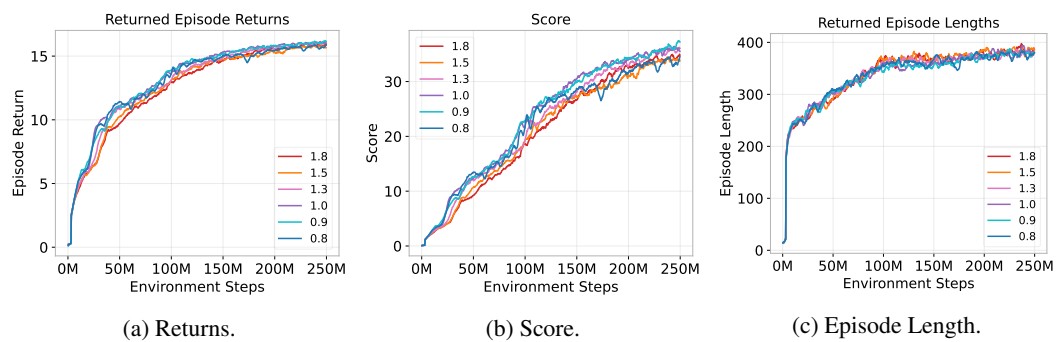

(a) Returns.
(b) Score.
(c) Episode Length.

Figure 17: Effect of the transient learning rate on performance.

| Parameter | Value |
|-----------|-------|
| **General Configuration** | |
| ALG_NAME | `pt_minhash` |
| NUM_MINIBATCHES | 4 |
| NUM_EPOCHS | 1 |
| NORM_INPUT | True |
| NORM_TYPE | `layer_norm` |
| LR | 0.001 |
| MAX_GRAD_NORM | 1.0 |
| **Transient Memory (IHT)** | |
| TRANSIENT_TABLE_SIZE | 128 |
| NUM_HASHES | 2 |
| NUM_ROWS | 128 |
| SLOTS_PER_BIN | 4 |
| TRANSIENT_LR | 0.5 |
| DECAY | 0.95 |
| $\rho$ | 0.75 |

Table 6: Hyperparameters for `PT_minhash` (IHT).

| Parameter | Value |
|---|---|
| TRANSIENT_TABLE_SIZE | 512 |
| NUM_HASHES | 2 |
| NUM_ROWS | 128 |
| TRANSIENT_LR | 0.5 |
| DECAY | 1.0 |

Table 7: Tranisent Memory Hyperparameters for PT-IHT (permanent retains from PQN).

| Parameter | Value |
|---|---|
| TRANSIENT_LR | 0.003 |
| DECAY | 1.0 |

Table 8: Hyperparameters for `PTQ-NeurIPS` (permanent retains from PQN).

| Parameter | Value |
|---|---|
| TRANSIENT_TABLE_SIZE | 128 |
| NUM_HASHES | 2 |
| NUM_ROWS | 128 |
| SLOTS_PER_BIN | 4 |
| TRANSIENT_LR | 0.5 |
| DECAY | 0.95 |
| $\rho$ | 0.75 |

Table 9: Hyperparameters for `TM-Only` ablation.

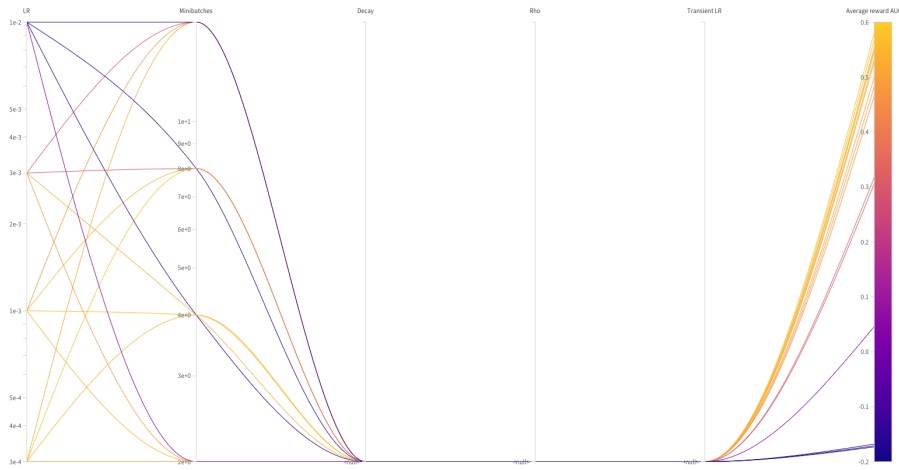

Figure 18: HP tuning plot for the PQN baseline.

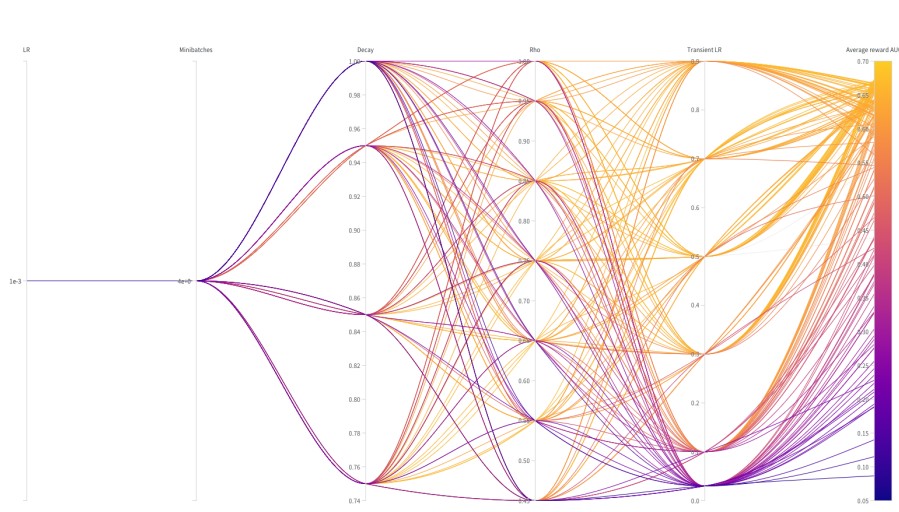

Figure 19: HP tuning plot for the PT MinHash method.

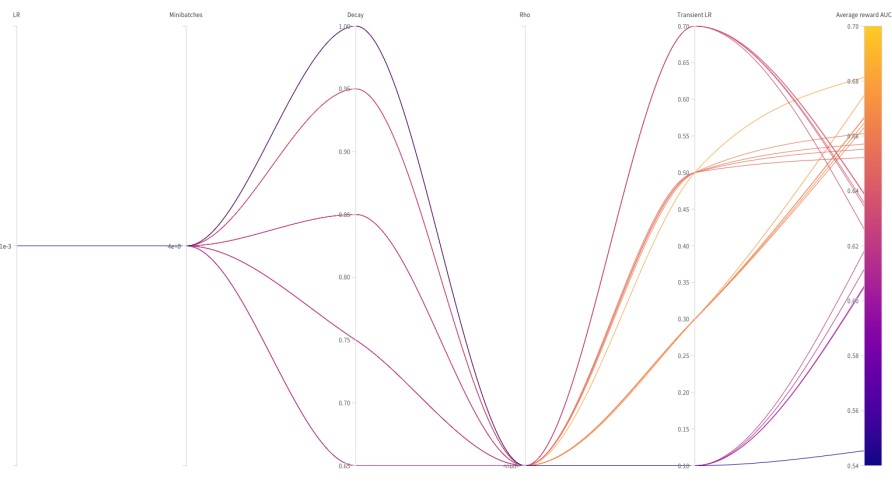

Figure 20: HP tuning plot for the PT-IHT method.

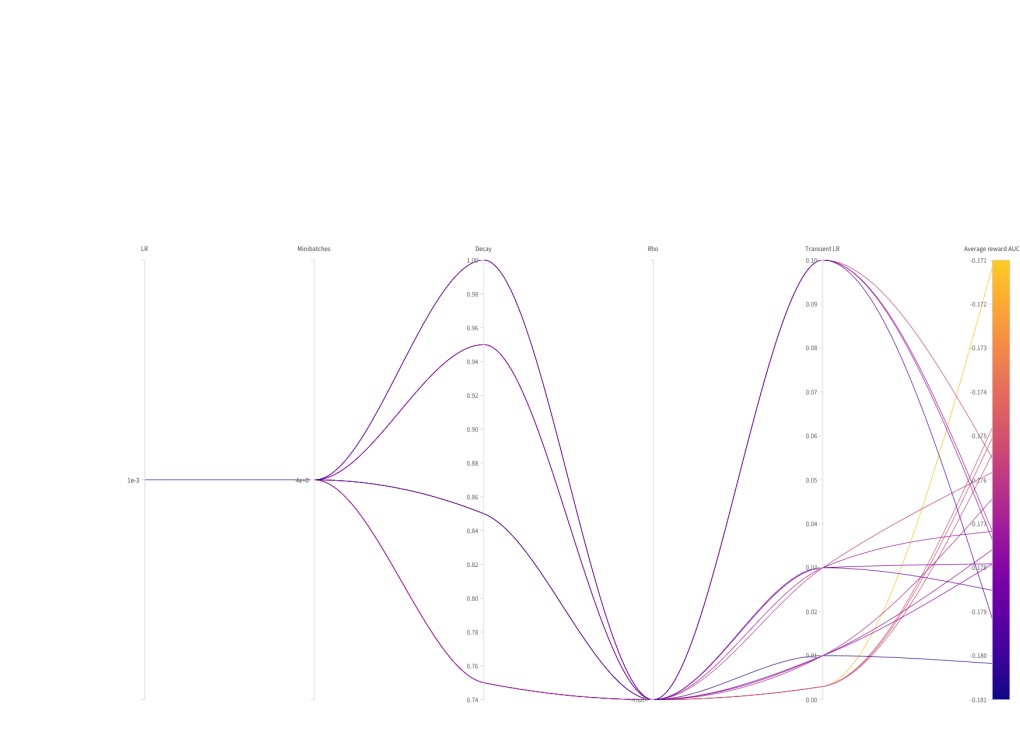

Figure 21: HP tuning plot for the PTQ-NeurIPS baseline.

