# OpenReview forum: "Permanent and Transient Representations for Continual Reinforcement Learning"
_ICLR.cc/2026/Conference — Submitted to ICLR 2026_

### Official Review · Reviewer_CwA7 · 2025-10-23

**Soundness:** 2
**Presentation:** 1
**Contribution:** 3
**Rating:** 4
**Confidence:** 4

**Summary:**

The paper builds on previous work in continual reinforcement learning, where decomposing an agent into permanent and transient components was proposed as a way to address the stability–plasticity trade-off.
It extends this idea by using different feature representations for these components. In particular, instead of using a neural network for the transient component, it introduces a feature representation method that tokenizes and hashes observations, and estimates and updates values like tile coding.
The paper evaluates this approach on non-stationary prediction, control (chain problem), and large-scale control (Craftax) problems, and shows improvements over the previous baseline.

**Strengths:**

1. The paper is well-motivated, the related literature review is covered, and the idea of separating permanent and transient feature representations is useful.
2. The paper also has experimental rigor in most of the cases, as it mentions numbers of seeds, details of hyperparameters, and enough details of the environments and experiment settings.
3. The paper is aware of its limitations. e.g., the semi-continual settings (agent is aware of the task boundaries), and parallelization of environments, which is not ideal

**Weaknesses:**

1. (line 269) The paper mentions feeding separate features to transient vs permanent modules in the control experiment. This hand-crafted separation can help the algorithm learn better, but ideally, a continual learner should be able to receive the raw stream of observations and attain the more important features for each component autonomously.

2. The feature representation method works with symbolic observations, as mentioned in the paper. How can the method be generalized to environments with other forms of observations?

3. The order of chapters should be improved. Before diving into the results, the method should be introduced. In particular, the order of chapters 5 and 6 should be changed.

4. The proposed idea comes with hyperparameters, like $\rho$. The paper would benefit if there were a paragraph about the effect of these hyperparameters and how they should be set for new environments and settings. (I see plots in the appendix on these hypers, but more explanation in a paragraph or two is needed in the main text)

5. The large-scale experiment can be improved. Three seeds are not enough to confidently compare the algorithms. Furthermore, it is valuable to compare PQN(1) and PQN(32), to show the effect of update frequency and batch size, but they are still the same algorithm with different hyperparameters compared. The paper would benefit if a separate plot were presented for hypers.

Minor comments:

(Line 475)- And should be lower case.

It would be better to keep the supplementary material at the end of the paper, not taken out of the paper, so that referring to the appendix for algorithms and hyperparameter details is easier.

**Questions:**

1- How will PT-TD(ours) perform if all features are fed both to the transient and permanent components?

2- How do you tune the hyperparameters specific to your method (e.g. p)?

3- How can you scale this representation method to other types of observation, like to images? Can you run some small experiments to show your method works for those settings as well?

4- The paper states the idea that the permanent features of the observations are captured in the permanent network, and the changing, transient characteristics of the environment, in the transient component. This is an assumption, which hypothetically will help with better performance, and measuring performance can be one way to move toward validating this statement. My question is, how can you actually measure what is stored in which component? E.g., can measuring similarities between the feature representations, and permanent observation features vs transient ones help? Are there diagnostic metrics (e.g., representational similarity analysis, interference measures) that could confirm the intended division between permanent and transient features?

5- Can you elaborate on what you mean by “append inventory, intrinsic values” in line 421?

6- Some assumptions, such as full rank for feature matrices, in the theoretical results are overly strong. How will the theorems change in case of violations of these assumptions?

---

> ### Author Response · Authors · 2025-11-26
>
> We thank the reviewer for their effort and time on the review. We address the critiques and questions below:
> * **"Hand-crafted" Features vs. Automatic Discovery:** While we used hand-crafted features for the small-scale experiments (Chain/Grid), our large-scale experiments (Craftax and Images) do not use hand-crafted separation. The permanent component autonomously learns features via a neural network.
>
>     To explicitly test "autonomous" feature discovery for the transient component, we implemented a fully neural PTQ-NeurIPS baseline (see General Response). As detailed there, this purely neural approach failed to learn in online settings due to update instability and slow initial learning. This confirms that while "autonomous" features are ideal, a non-parametric structure is necessary for the rapid adaptation component in this setting.
>
> * **Generalized to environments with non-symbolic observations:** Following your comment, we have included a new, fully online, image-based experiment to demonstrate generalizability. Please see Section 8 in the updated draft and our General Response for details. The preliminary results demonstrate that our method adapts significantly faster than baselines in a pixel-based environment.
>
>
> * **Seeds:** As outlined in the general response, we increased the number of seeds to 10 for the 250M variant (results remain consistent), and all other experiments (including the new Image task) use 30 seeds (reporting 90% CIs).
>
>
> * **Order of chapters:** Thanks for suggesting. We believe the presentation is a personal taste (other reviewers appreciated the current presentation), and would like to keep it this way.
>
> * **Paragraph on hyperparameters:** We appreciate the suggestion to explain the hyperparameters' effects. We have added a detailed section in the Appendix. To summarize the key intuitions:
>     * *Tables (T):* Similar to tiles in Tile Coding. More tables spread information, increasing generalization. Fewer tables concentrate information, reducing generalization.
>     * *Slots (S):* Determines capacity. Too few slots lead to high eviction rates; too many slots approach tabular memory.
>     * *Generalization ($\rho$):* Explicitly controls the mixing of values between slots. $\rho$=1 forces isolation (no generalization); $\rho$<1 enables local smoothing.
>     * *Number of Hashes (K):* Determines the precision of the MinHash signature. A higher K better preserves the Jaccard similarity property (improving retrieval accuracy), though with diminishing returns beyond a certain point.
>
> * **HPs:** The hyperparameters are problem-dependent (like learning rate), and should be tuned for each continual RL environment. We have added these details to the Appendix and also summarize them below. We expect meta-learning approaches to be helpful in automatically tuning them, which could be an interesting future direction.
>     * *Craftax Baselines:* For the baselines and the permanent network in all variants of PTQ-learning, we used hyperparameters from the PQN repository, which are consistent with the results published in the PQN paper. We only tuned for the number of minibatches for the PQN baseline, because it determined the number of steps the permanent network would take, which we also tuned for our approach. Transient parameters were found by performing a search over a reasonable range of values.
>     * *Craftax:* Given the large hyperparameter space, we used coordinate-wise search: we fixed all hyperparameters but one to a reasonable baseline and performed a grid search for that single hyperparameter. This process was then repeated iteratively for each hyperparameter, fixing the newly tuned value before moving to the next. For each configuration, we ran a total of 150M steps with 1024 environments in parallel, using AUC as the selection metric. The final results were reported for 250M steps. We used the same HPs for the online craftax experiment (minibatch was reduced to 8 to fit the smaller batch size).
>     * *Image Tasks:* For the image tasks, we performed a grid search over all the hyperparameters for the 256-image variant. Once we found the best values, we fixed them and tested the performance for the 500- and 1000-image variants.

---

> > ### Author Response · Authors · 2025-11-26
> >
> > * **Q1.** All baselines receive the full set of features. The benefit of our approach comes from explicitly decoupling the representations. This reduces interference: the rapid, high-variance updates of the transient component are prevented from destabilizing the slow, stable features of the permanent component.
> >
> > * **Q4. Desiderata of permanent and transient features:** Our choices are grounded in CLS theory (Kumaran et al., 2016). We use a Neural Network for the permanent component to capture slow, generalized structure (Neocortex-like). We use a non-parametric approximator for the transient component to enable precise, rapid corrections without interference (Hippocampus-like).
> >
> > * **Q5: “append inventory, intrinsic values”:** This refers to concatenating the inventory vector with the observation vector before hashing, to ensure the hash signature captures the full state (agent view + intrinsics + inventory).
> >
> > * **Q6. Some theoretical assumptions are overly strong:** All the assumptions we made in theory, including a full column rank feature matrix, follow standard results in the linear setting (Borkar 1997; Tsitsiklis & Van Roy 1996; Sutton et. al., 2015).

---

### Official Review · Reviewer_QcCf · 2025-10-30

**Soundness:** 2
**Presentation:** 3
**Contribution:** 2
**Rating:** 4
**Confidence:** 4

**Summary:**

The manuscript investigates the stability–plasticity trade-off in continual reinforcement learning (CRL) by decomposing value estimation into permanent and transient components with explicitly separate feature representations. The central idea is that permanent value functions should leverage slow, stable features, while transient value functions should use fast-evolving or non-parametric, reward-predictive features to adapt rapidly. To enable rapid adaptation at scale, the manuscript proposes a non-parametric transient memory based on MinHash signatures that preserve Jaccard similarity for symbolic observations, providing tabular-like fast updates with controllable local generalization. Theoretical analyses and small-scale experiments are presented to motivate the design, and larger-scale results on Craftax-Classic show improvements over a PQN baseline.

**Strengths:**

1. Clear motivation and organization: from intuition to theory, small-scale validations, and finally large-scale experiments.
2. Decoupling the representations used by permanent and transient value functions is a simple and compelling idea for improving the stability–plasticity balance.
3. The manuscript includes theoretical results, algorithmic details, and pseudocode, which improves transparency and reproducibility.
4. The method shows improvements on large-scale experiments over the PQN and demonstrates applicability.
5. The proposed framework for separating representations of permanent and transient components may be useful beyond the presented setting, and the memory design could transfer to other CRL scenarios.

**Weaknesses:**

1. The core advance over PT-TD is primarily the use of separate feature sets for the two value components; while intuitive and useful, it may be viewed as a modest extension rather than a substantial methodological leap.
2. The work has significant limitations as it does not extend the applicability of the original PT-TD method (such as how to apply it to other policy based reinforcement learning algorithms)
3. The large-scale evaluation compares mostly to PQN and omits direct comparisons to PT-TD and other CRL approaches, making it difficult to attribute gains to the proposed decomposition and memory design.
4. It is not fully clear how the theoretical results directly support the central empirical claims. Clearer mapping from lemmas/theorems to observed phenomena would strengthen the argument.
5. The manuscript lacks ablation studies and visual analysis.
6. Reporting and clarity issues: notation is introduced (V, Q, s, etc.) without sufficient definitions; some figure elements (e.g., the dashed line in Fig. 2(a)) and experimental variants (e.g., “TM-only”) are not clearly explained.

**Questions:**

1. In Section 5.2, the transient component uses illumination features that are directly reward-predictive, while the permanent component uses RGB features. If baselines receive all features, does this create a mismatch in inductive bias that favors the proposed method?
2. If the observation similarity assumption (needed for MinHash/Jaccard) does not hold, does the transient memory still function effectively?
3. Why is PQN the sole large-scale baseline, especially given PQN’s design goal of leveraging environment parallelization (line 436)? Please add comparisons to PT-TD and representative CRL baselines (e.g., EWC/L2 regularization for value networks, rehearsal/buffering, meta-learning-based CRL) or justify in detail why they are not included.

---

> ### Author Response · Authors · 2025-11-26
>
> We thank the reviewer for their effort and time on the review. We address the critiques and questions below:
> * **Modest extension; extension to policy-based reinforcement learning algorithms:** While Anand & Precup (2023) introduced the conceptual PT-framework, they relied on shared representations, which fundamentally limited the system's ability to trade off stability and plasticity. We note that update rules alone are insufficient to fully realize complementary learning systems computationally, and in this work, we present a strong case for leveraging distinct representations to separate the slowly learned general predictions from the rapidly adapting transient component. As such, our work:
>     * Presents the first empirical/theoretical evidence for using distinct representations (vs. shared features).
>     * Analyzes the framework theoretically in the linear function approximation setting using standard assumptions (first to do so, as previous work was tabular);
>     * Introduces a novel non-parametric approximator, which generalizes Tile Coding, to effectively learn transient updates on complex environments (previous work used neural networks);
>     * Demonstrates the effectiveness of our approach at scale, on the complex Craftax environment, and the newly added image task (previous work had small-scale experiments).
>
>     We agree that extending this to policy-gradient methods would be interesting, but we restrict our scope here to value-based RL. Note that for actor-critic methods, the value function estimation could be done in the way we propose here in a straightforward way.
>
> * **Baselines:** Following your suggestion, we have included a PTQ-NeurIPS baseline. As detailed in the general response, this approach failed to learn in the fully online settings (Image MDP and 2-parallel envs) due to update instability and the slow initial learning typical for gradient-based methods. This result confirms that non-parametric, controlled generalization is necessary for the rapid adaptation required in continual RL.
>
>     We did not directly compare our approach with EWC or meta-approaches because they are orthogonal and complementary to our contribution. EWC acts as a regularizer on the permanent network (optimizer level), while our method is a decomposition of the value function. One could combine them (e.g., using EWC to update the permanent value network) to further improve performance, or meta-learn the transient hyperparameters for faster adaptation. We have now included this point in the discussion section.
>
> * **Ablations:** We performed extensive parameter-based ablations on the Craftax 250M benchmark and the newly added image experiments (see Appendix Figures 10-15). Additionally, the TM-only baseline and the newly added IHT-based baseline are ablations of our method that validate our design choice (see the general comment for details).
>
> * **Practical impact of theory:** We appreciate your comment on the theory. We view our convergence proof (under standard assumptions) as providing rigour and a necessary theoretical foundation to justify our approach, analogous to how convexity assumptions justify SGD/ADAM, even if they are frequently used in scenarios where those assumptions are not satisfied (e.g., neural networks). Theory confirms our approach is mathematically sound.
>
> * **Q1. If baselines receive all features[...]?** All baselines receive the full set of features. The benefit of our approach comes from explicitly decoupling the representations. This reduces interference: the rapid, high-variance updates of the transient component are prevented from destabilizing the slow, stable features of the permanent component.
>
> * **Q2. [...]observation similarity assumption[...]?** Preserving Jaccard distance is a property of MinHash; it is not an assumption on observations. Intuitively, if two observations share many tokens, they will share many hash bits and map to the same bins. This allows the RL agent to generalize to semantically similar states naturally.

---

### Official Review · Reviewer_Joyf · 2025-10-31

**Soundness:** 2
**Presentation:** 2
**Contribution:** 1
**Rating:** 2
**Confidence:** 3

**Summary:**

The paper proposes to extend the **Permanent-Transient** (PT) framework (Anand & Precup 2023) by introducing separate feature representations for the permanent and transient components of the value function. Additionally, the authors introduce a non-parametric transient memory based on MinHash and CMAC-like updates to enable rapid, online adaptation. The method is evaluated on small-scale toy problems and the Craftax-Classic environment, claiming improved performance over PQN baselines.

**Strengths:**

* The idea of separating permanent and transient representations is biologically and intuitively motivated.
* Provides theoretical convergence proofs under linear approximation (although very classical).
* Introduces a novel non-parametric memory module that could be reusable beyond this setting.

**Weaknesses:**

* The claimed novelty (“separate feature representations”) feels incremental over Anand & Precup (2023). Why is this a significant step rather than a straightforward design variant?
* The "transient memory" resembles *tile coding* with *hashing*, which is decades old. The description sounds more like a new combination of existing mechanisms rather than a fundamentally new idea.
* The paper leans heavily on biological analogies (CLS theory) without providing an actual mechanistic link or empirical justification beyond metaphor.
* The convergence result essentially repackages standard two-timescale SA arguments (Borkar 1997; Tsitsiklis & Van Roy 1996). What exactly is new here beyond the trivial extension to two feature matrices?
* No discussion of how feature misalignment or correlation between permanent and transient features affects convergence speed or stability.
* The assumptions (e.g., observable task boundaries, full column rank of feature matrices) are unrealistic in genuine CRL scenarios. How would the analysis hold when boundaries are unknown or tasks are not i.i.d.?
* The toy tasks are far too simple to substantiate the claims. Gridworld and chain environments are saturated benchmarks for incremental TD methods; they say little about continual learning dynamics.
* No measure of catastrophic forgetting or transfer between tasks; all reported metrics are single-task reward curves, so the continual aspect remains speculative.
* Code release is promised “upon acceptance”, which should be available at submission time.
* Heavy use of ChatGPT/Copilot is acknowledged; this raises questions about authorship and whether critical sections were written or validated by the authors themselves.

**Questions:**

Questions are given under the Weaknesses.

---

> ### Author Response · Authors · 2025-11-26
>
> We thank the reviewer for their review. We address below some of the points raised in their review:
> * **Novelty:** We respectfully disagree that our work is incremental. Although Anand & Precup, 2023 introduced the PT-framework and learning algorithms, they relied on shared representation, which limited the agent’s ability to trade off stability and plasticity.
> Also, as noted by the reviewer, another goal is to fully realize the desiderata of complementary learning systems theory computationally. Update rules alone are insufficient, and in this work, we present a strong case for leveraging distinct representations to separate the slowly learned general predictions from the rapidly adapting transient component. As such, our work:
>     * Presents empirical and theoretical evidence for using distinct representations (previous work introduced update rules only);
>     * Analyzes the framework theoretically in the linear function approximation setting using standard assumptions, which is more general and complex (previous work was tabular);
>     * Introduces a novel non-parametric approximator, which generalizes Tile Coding, to effectively learn transient updates on complex environments (previous work used neural networks);
>     * Demonstrates the effectiveness of our approach at scale, on the complex Craftax environment, and the newly added image task (previous work had small-to-medium-scale experiments).
> * **Resemblece with tile coding:** The reviewer is right, and we view this as a strength. Tile coding, due to the curse of dimensionality, was limited to problems with low-dimensional environments. Consequently, hashing-based tile coding is commonly used. However, to the best of our knowledge, MinHash hashing-based tile coding has not been explored. We implemented this baseline (see General Response), but this simple IHT-based approach results in undesirable generalization due to hash collisions resulting in slower adaptation. Our non-parametric approximator, because of slot-based storage, is more general, and it is well-suited to estimate the transient value function on complex continual RL domains with symbolic and image observations. Please see the general comment that validates our design choices.
> * **Theoretical assumptions; convergence:** To the best of our knowledge, all linear function approximation results follow the standard proof templates described in Tsitsiklis & Van Roy (1996) for single-timescale methods, and Borkar (1997) for two-timescale algorithms. Our goal was to analyze our specific framework within this established linear result, not to invent new optimization theory. Thus, we adopted the standard assumptions and proof techniques widely accepted in the RL literature. Our theory is an application of these tools to justify the use of distinct representations.
>
>     _Regarding feature misalignment:_ These standard two-timescale results ensure asymptotic stability and convergence under standard assumptions (full column rank). Because the slow process appears stationary to the fast process, and the fast process appears converged to the slow process, stability is guaranteed regardless of specific feature correlations or misalignment. **Indeed, Corollary 1 establishes convergence even in the case of identical features (full correlation), confirming that our stability results hold in a fully correlated setting.**
>
>     We acknowledge that task boundary and i.i.d. task-sampling assumptions are simplifications, but they are necessary for tractable theoretical analysis (as in prior works). Our large-scale experiments demonstrate the method's practical success in complex continual learning environments where these assumptions are relaxed.
> * **Toy tasks only:** We present toy tasks (chain and grid environments) to motivate the need for distinct representations. **Our main empirical result is presented in Section 7 on the complex Crafter environment with two variations (250M and online). We also added image-based tasks (see the general comment for details).**

---

> ### Author Response · Authors · 2025-11-26
>
> * **Continual aspect remains speculative:** As outlined in line 66, we view continual learning as endless adaptation following Abel et al., 2023 and Sutton et al., 2007, where the goal is to maximize rewards from an endless non-stationary stream of experience, emphasizing the agent’s ability to adapt and learn forever. This is in contrast to transfer learning, for example, where forgetting is measured explicitly because task boundaries are available. However,  note that **we have already included the performance on tasks other than the online task for the chain and grid tasks, where the task boundaries are visible to the agent (see Section 5, Figure 2).**
> * **Code release; LLM-usage:** We adhere to ICLR policies in both regards. We are happy to provide the code to the AC upon request for verification and will only release it upon the paper acceptance.
> Regarding LLM usage, we must state clearly: All core ideas, scientific claims, theoretical analyses, and experimental designs are entirely our own. Our acknowledgment was made in the spirit of transparency following ICLR guidelines, and it refers to standard use, such as code completion, grammar corrections, and polishing. These tools had no bearing on the scientific contributions of this work.

---

### Official Review · Reviewer_MWuz · 2025-11-03

**Soundness:** 3
**Presentation:** 3
**Contribution:** 2
**Rating:** 4
**Confidence:** 3

**Summary:**

This paper builds upon the Permanent-Transient (PT) value function decomposition framework for Continual Reinforcement Learning (CRL). The core idea is to use separate feature representations for the permanent (slow, stable) and transient (fast, plastic) value functions, arguing that this better embodies the complementary learning systems inspiration. The authors provide theoretical convergence guarantees for this setup and introduce a novel non-parametric approximator for the transient value function, which uses hashing and a slot-based memory to enable fast, online learning with controlled generalization. The method is evaluated on small-scale prediction/control tasks and the large-scale Craftax-Classic benchmark, where it is shown to outperform the PQN baseline, especially with a small number of parallel environments.

**Strengths:**

- **Well-Motivated Core Idea**: The extension of the PT framework to use separate representations is intuitive and well-justified from a biological and functional perspective. It is a logical and meaningful step forward from prior work.
- **Theoretical Analysis**: The paper provides a solid theoretical foundation, establishing convergence guarantees for the proposed method under linear function approximation (where previous works focused on tabular settings) using a two-timescale analysis. This rigor is a significant strength.
- **Novel Non-Parametric Transient Memory**: The proposed non-parametric approximator is a creative and interesting contribution. Its design, combining ideas from tile coding, CMACs, and MinHash, is well-explained and appears effective for the environments tested.
- **Empirical Success:** The method demonstrates strong empirical performance, convincingly outperforming the PQN baseline in both small-scale experiments and on the challenging Craftax benchmark.

**Weaknesses:**

- **Limited Scope and Generalizability of Transient Memory**: The non-parametric transient memory is currently limited to symbolic, grid-like observations. The authors briefly mention that randomly initialized CNNs could be used for RGB images (line 470). Given that many important CRL domains (e.g., Atari, real-world sensors) involve high-dimensional pixel inputs, this is a significant limitation. The paper would be substantially stronger with an experiment demonstrating the method's applicability beyond symbolic domains.
- **Lack of Ablation and Comparison to Related Memory Approaches:** The paper misses an opportunity to properly situate its novel transient memory within the existing literature.
- **There is no direct comparison to classic function approximators like CMACs or Tile Coding**, which share similar intuitions about local generalization. Showing that the proposed method is superior to or offers unique advantages over these established techniques would strengthen the contribution.
- **The connection to episodic memory methods in RL is very relevant but unexplored**. A discussion or comparison would help readers understand the relationship between this work and other memory-based approaches.
- **Limitation of Transient Memory's Differentiability**: The non-parametric nature of the memory makes it non-differentiable, which may limit its integration with end-to-end learning systems. The authors do not discuss potential pathways to a differentiable variant, which would be a valuable direction to mention in the discussion. A possible variant can be episodic memories.
- **Unconvincing and Potentially Self-Contradictory Online Experiment Setup:** The definition and setup of the "online" experiment are confusing and weaken the paper's narrative.
  - The authors state that "an ideal, general-purpose CRL algorithm should learn effectively from a single stream of experience" (line 390). However, their "online" experiment uses two parallel environments. This seems to contradict their own stated ideal. The community standard for "online" RL is typically a single environment. The choice to use two environments without a compelling justification (e.g., a comparison to a true single-environment run) undermines the claim of online learning capability.
  - Suggestion: The authors should either run an experiment with a single environment (updating the permanent network with a batch size of 1 or with a small batch collected sequentially, like 32) to truly validate their online learning claim, or they should reframe their "online" experiment as a "low-parallelism" setting and adjust their narrative accordingly.



This paper presents a valuable extension to the PT framework with a novel and effective non-parametric memory architecture. The theoretical analysis and strong empirical results on Craftax are commendable. However, the current version has significant weaknesses that prevent a higher score. The contribution would be substantially greater if the method's applicability to broader domains, like pixel-based observations, were demonstrated. Furthermore, a more rigorous comparison to existing memory-based approximators is needed.

I lean towards rejection in its current form, primarily due to the limited scope of the transient memory. However, if the authors can address the major concerns, especially by adding experiments that demonstrate generalizability beyond symbolic inputs and add necessary ablation and baseline comparisons, I'll increase my score.

**Questions:**

- Online Experiment Justification: Why was the decision made to use 2 environments in the "online" experiment instead of a single environment, especially given the stated ideal of learning from a "single stream of experience"? Can you provide results with a single environment to solidify this claim?
- Generalizability: Can you provide any empirical evidence (even preliminary) that your hashing-based transient memory can be adapted to non-symbolic observation spaces like RGB images, for instance, using a random CNN projection as mentioned?
- Baseline Comparisons: How does your non-parametric transient memory compare, in terms of performance and efficiency, to a simpler implementation using standard Tile Coding or a CMAC on the Craftax benchmark?
- There is no ablation for the transit memory component system, which makes it hard to assess. Can the authors provide an ablation that support the design choices and coponents of that transient memory system?
- Memory Consolidation: The paper leaves "when and what to consolidate" as future work. Did you experiment with any simple consolidation strategies (e.g., moving values from transient to permanent memory)?

---

> ### Author Response · Authors · 2025-11-26
>
> We thank the reviewer for their thorough and insightful feedback. We appreciate the many connections mentioned in their review, and we address their critiques and questions below.
>
>
> * **Limited Scope and Generalizability of Transient Memory:** Following your recommendation, we have included a new, fully online, image-based experiment to demonstrate generalizability. Please see Section 8 in the updated draft and our General Response for details. The preliminary results demonstrate that our method adapts significantly faster than baselines in a pixel-based environment.
> * **Direct comparison to CMACs and tile coding:** Note that Tile coding requires low-dimensional continuous vector observations, as it suffers from the curse of dimensionality, limiting its use in large, complex domains. Consequently, hashing-based tile coding is commonly used as an alternative. However, to the best of our knowledge, MinHash hashing-based tile coding has not been explored. We have now included a MinHash-IHT baseline, which is a simplified version of our method that mimics classic Tile Coding (direct mapping to bins without slots/LRU). Our experiments show that while IHT works on simple tasks, its performance deteriorated significantly when the size of the observation space increased due to hash collisions, validating that our specific design choices (controlled generalization and slot-based storage) are essential for scaling to complex domains (see updated draft and the main comment for details). Also, to the best of our knowledge, MinHash hashing-based tile coding has not been explored before.
> * **Ablations:** We have performed extensive parameter-based ablation on the Craftax 250M benchmark and the image experiments (see Appendix). The TM-only baseline and the newly added IHT-based baseline are ablations of our method that validate our design choice (see the general comment for details).
> * **Episodic memory:** We thank the reviewer for this insightful pointer. We have added a paragraph in the discussion section in the updated paper clarifying the distinction:
> > While both approaches use key-value storage, Episodic Memory typically acts as a non-parametric replay buffer that stores and retrieves specific past returns (or Q-value estimates) via complex kernel regression (averaging neighbours). In contrast, our non-parametric component is a function approximator. The values stored in our hash table are residuals, learned and updated via TD-error using simple summation. Consequently, our approach is designed for rapid adaptation in continual RL, rather than to accelerate single-task convergence. As the reviewer correctly noted, our method is closer in spirit to tile coding rather than episodic retrieval.
> * **End-to-end differentiability:** We have included a fully neural network-based baseline (PTQ-NeurIPS; see General Response). This approach failed to learn in the fully online settings (Image MDP and 2-parallel envs) due to update instability and the slow initial learning, a characteristic of gradient-based methods. This result confirms that non-parametric, fast updates are necessary for the rapid adaptation required in continual RL.
> * **Online Experiment Setup:** We clarify that our method is fully online: the transient values (and thus the overall value estimates) are updated at every timestep using the immediate TD-error. While compute constraints limited the length of the single-environment Craftax run, we addressed your concern directly in the new Image MDP experiment. This experiment uses a single environment in an online setup, where our method demonstrated superior performance and stability compared to baselines. We have added the "low parallelism" note regarding Craftax to the footnote for clarity.
> * **Consolidation:** We currently use the strategy from Anand & Precup (2023): the permanent component is updated in phases using data collected during that phase. Biological intelligences consolidate selectively during sleep; identifying which experiences are "important" to consolidate remains an open research question. From a practical standpoint, our current phase-based approach ensures the transient component has sufficient time to perform meaningful corrections before those corrections are poured into the permanent network.

---

### Author Response · Authors · 2025-11-26
**General Response**

We thank all reviewers for their time and thoughtful feedback. We are encouraged that the reviewers found our core idea "well-motivated," "intuitive," and "rigorous". As several common themes emerged in the reviews, we first provide a summary of our major updates—including new experimental results—that directly address these points.


* **Generalizability of Transient Memory beyond symbolic observations (MWuz and CwA7):**
In response to the reviewers' request to demonstrate generalizability beyond symbolic domains, we conducted a **fully online, image-based experiment (see Section 8)**. Our experiments revealed the general applicability of our MinHash-based non-parametric function approximator. In particular, our MinHash-based PTQ-learning demonstrated significantly faster adaptation compared to other approaches when the tasks changed, resulting in higher cumulative rewards. We also tested a simplified Minhash-IHT-based PTQ-learning (details of the method in the next bullet point). While this approach performed well in small observation spaces, its performance deteriorated in larger spaces due to uncontrolled generalization and higher collision rates. This confirms that our method's controlled generalization is critical for scaling to complex, pixel-based environments.

    To obtain tokens to compute MinHash signatures, we leveraged a pre-trained convolutional neural network. This setup allows us to isolate and demonstrate the core contribution: the transient memory's ability to adapt instantly. As discussed in Section 10 of the updated draft, this requirement can be relaxed in future work by:
    * Leveraging a pretrained vision encoder or traditional CV techniques (bag of visual words);
    * Adapting deep hashing to bypass the tokenization step and directly compute a hash signature;
    * Exploiting the inductive biases of randomly initialized CNNs along with a small, trained projection layer to obtain the token vector;


* **Experiments:**
    * **Ablations (MWuz and QcCf):**
        * *Parameter Robustness:* We’d like to highlight that extensive parameter sensitivity analysis on the craftax benchmark is in the Appendix (Figures 10-15), which demonstrated our method's stability across a wide range of hyperparameter configurations. We have also confirmed this robustness on the new image-based task, whose main results can be found in Section 8, and hyperparameter results can be found in the Appendix.
        * *Component Necessity (MinHash-IHT):* To test the necessity of our design (slots and $\rho$), we implemented a simplified “MinHash-IHT” baseline that mimics classic Tile Coding (direct mapping to bins without slots). While effective on simple tasks, this baseline’s performance deteriorated significantly on complex, large observation space tasks (e.g., the Image MDP) due to uncontrolled generalization resulting from hash collisions. This empirically validates that our method's slot-based value estimation with controlled generalization ($\rho$) is essential for scaling to complex environments. (Note: We also retain the TM-only baseline to validate the permanent component.)
    * **Neural Baselines & Differentiability (Reviewers MWuz, QcCf, and CwA7):** To address requests for end-to-end differentiability and automatic feature discovery, we implemented a fully neural PTQ-NeurIPS baseline. While this neural baseline learns meaningful predictions in the large-batch setting (250M benchmark), it fails to learn in the fully online, low-batch settings (Image MDP and 2-parallel envs) due to update instability. Moreover, it suffers from the slow initial learning typical of gradient-based methods. This result confirms that a purely neural approach struggles with the rapid updates required for continual RL, thereby justifying the necessity of our non-parametric approximator for stable, online, and rapid adaptation.
    * **Experimental Rigour & Seeds (Reviewer CwA7):** the 250M Craftax benchmark has been updated to 10 seeds from 3 seeds, with results remaining consistent. All other experiments, including the new image-based task, were conducted with 30 seeds, and we report the mean and 90% confidence interval (z=1.645). (Note: The online Craftax experiment (2-parallel envs) retains 3 seeds due to the significant computational runtime (18h+), but its trend aligns with the higher-seed experiments.)

---

### Meta-Review · Area_Chair_oF2s · 2026-01-04

**Summary:**

This paper proposes a method for continual reinforcement learning (CRL) that separates an agent's representation into two components: a "permanent" component for stable knowledge retention and a "transient" component for rapid adaptation to new information. The work extends the Permanent-Transient (PT) framework of Anand & Precup (2023) by using a non-parametric feature representation for the transient value function. The authors provide a theoretical convergence proof under linear function approximation and present experiments to demonstrate their method's effectiveness.

The primary concern from the reviewers is the limited novelty. This lack of novelty is compounded by weak and unconvincing empirical validation. Finally, the theoretical analysis, while present, is not a strong point of support due to its reliance on idealized assumptions that are disconnected from the practical challenges of continual RL. The poor presentation further hinders the paper's ability to convey its ideas effectively. The author's rebuttal did not adequately resolve these major issues. Therefore, I recommend rejection.

**Reviewer Concerns:**

* *Incremental Contribution:* All reviewers found the work to be a minor and incremental extension of existing work, primarily Anand & Precup (2023). The core methodological contribution -- using a non-parametric approximator for the transient part -- was not seen as a significant leap. This has not been addressed by the rebuttal.

* *Weak Empirical Evidence:* The experimental validation was found to be insufficient by all reviewers. The toy tasks were deemed too simplistic to support the paper's claims. More importantly, the main large-scale experiment on Crafter was criticized for lacking comparisons to the most crucial baselines. This has been partially clarified and addressed by the rebuttal.

* *Unrealistic Assumptions:* While the paper provides a convergence proof, the reviewers pointed out that it relies on overly strong and unrealistic assumptions (e.g., observable task boundaries) that do not hold in genuine, challenging CRL settings. This limits the relevance of the theoretical results.

**Reviewer Scores:**

Given the fundamental nature of the reviewers' criticisms and the failure of the rebuttal to address them, it is highly unlikely that a full discussion period would have changed the outcome. The scores of 2, 4, and 4 may likely have been maintained.

---

### Decision · Program_Chairs · 2026-01-26

Reject